

# Multi-stage hybrid flow shop scheduling problem with lag, unloading, and transportation times

Lotfi Hidri and Mehdi Tlija

Department of Industrial Engineering, College of Engineering, King Saud University, Riyadh, Saudi Arabia

## ABSTRACT

This study aims to address a variant of the hybrid flow shop problem by simultaneously integrating lag times, unloading times, and transportation times, with the goal of minimizing the maximum completion time, or makespan. With applications in image processing, manufacturing, and industrial environments, this problem presents significant theoretical challenges, being classified as NP-hard. Notably, the problem demonstrates a notable symmetry property, resulting in a symmetric problem formulation where both the scheduling problem and its symmetric counterpart share the same optimal solution. To improve solution quality, all proposed procedures are extended to the symmetric problem. This research pioneers the consideration of the hybrid flow shop scheduling problem with simultaneous attention to lag, unloading, and transportation times, building upon a comprehensive review of existing literature. A two-phase heuristic is introduced as a solution to this complex problem, involving iterative solving of parallel machine scheduling problems. This approach decomposes the problem into manageable sub-problems, facilitating focused and efficient resolution. The efficient solving of sub-problems using the developed heuristic yields satisfactory near-optimal solutions. Additionally, two new lower bounds are proposed, derived from estimating minimum idle time within each stage *via* solving a polynomial parallel machine problem aimed at minimizing total flow time. These lower bounds serve to evaluate the performance of the developed two-phase heuristic, over measuring the relative gap. Extensive experimental studies on benchmark test problems of varying sizes demonstrate the effectiveness of the proposed approaches. All test problems are efficiently solved within reasonable timeframes, indicating practicality and efficiency. The proposed methods exhibit an average computational time of 8.93 seconds and an average gap of 2.75%. These computational results underscore the efficacy and potential applicability of the proposed approaches in real-world scenarios, providing valuable insights and paving the way for further research and practical implementations in hybrid flow shop scheduling.

Corresponding author
Lotfi Hidri, lhidri@ksu.edu.sa

# INTRODUCTION

The hybrid flow shop (HFS) is an industrial system consisting of sequential production stages, each comprising multiple identical parallel machines. In order to meet the criteria of being classified as an HFS, it is mandatory for at least one stage to contain more than one machine. The fundamental nature of an HFS lies in its ability to process a series of jobs in a specific order, flowing from the initial stage to the final stage. In this system, each machine within a stage can handle only one job at a time, and each job is exclusively processed by a single machine in each stage.

The HFS represents an extension of other shop configurations, including single machine, parallel machine, and flow shop. The scheduling problem within the HFS environment poses an intriguing challenge encountered in various real-world contexts, notably within manufacturing industries such as electronic ships (*Jin et al., 2002*), cable production (*Narastmhan & Panwalkar, 1984*), textile manufacturing (*Elmaghraby & Karnoub, 1997*), steel industry (*Jiang et al., 2023*), and glass manufacturing systems (*Geng & Li, 2023*). However, it is worth noting that the HFS scheduling problem, along with its different variants, has been proven to be NP-hard, presenting an important theoretical challenge (*Ruiz & Vázquez-Rodríguez, 2010*).

The scheduling problem related to HFS has been the subject of extensive research, resulting in a large body of available literature on the topic (*Hidri & Gharbi, 2017*; *Ribas, Leisten & Framiñan, 2010*; *Ruiz & Vázquez-Rodríguez, 2010*; *Wu & Cao, 2022*). The problem has been the focus of numerous studies aiming to solve it optimally or approximately using various algorithms, including heuristics, metaheuristics, and exact procedures. For further information on the HFS scheduling problems, readers are encouraged to refer to the cited literature (*Lee & Loong, 2019*; *Ribas, Leisten & Framiñan, 2010*; *Ruiz & Vázquez-Rodríguez, 2010*; *Tosun, Marichelvam & Tosun, 2020*).

Accurately modeling real-life situations in the context of the HFS scheduling problem necessitates considering parameters and constraints that have significant impacts, such as job release times, setup times, and machine unavailability, among others. Incorporating these parameters, including release dates and setup times, based on actual conditions can help bridge the gap between theoretical and practical aspects.

The unloading time, which refers to the duration required to remove a completed job from a machine, holds considerable importance in various industrial and manufacturing settings. It is a crucial real-life parameter that cannot be ignored, especially in specific applications such as sand casting (*Li et al., 2023b*). It becomes necessary to consider the unloading time separately from the processing time under certain circumstances when it becomes comparable to the processing time or more important. Steel industries commonly experience this problem, where parts removed from molds take longer to unload (remove) than the actual processing time. Furthermore, the bio-process industry, specifically in the context of utilizing fermentation techniques, encounters restrictions related to unloading time (*Gicquel et al., 2012*).

It is common in many HFS environments to ignore or neglect the transportation time between stages, when compared with the processing time. This scenario is frequently

encountered, particularly in cases where the distance between stages is neglected (*Shao, Shao & Pi, 2020*). However, in numerous real-life cases, the transportation time of a job becomes a pivotal factor that cannot be overlooked. In this context, researchers in *Amirteimoori et al. (2022)* tackled an HFS problem that incorporated transportation time considerations. They proposed a MILP (Mixed Integer Linear Programming) model. Due to the NP-hard nature of the problem under investigation, the authors proposed a metaheuristic approach called Particle Swarm Optimization-Genetic Algorithm (PSO-GA). The combination of the Particle PSO-GA metaheuristic was employed to achieve a solution that is close to optimal for the problem at hand. An experimental study was conducted to evaluate the proposed procedures, and the obtained results demonstrate the efficiency of these approaches.

Lag time in scheduling refers to the minimum time delay between the completion of a task or job and the start of a subsequent task or job. Lag time can occur due to various reasons, such as cooling (steel making industry), fermentation (bioprocess), or any other type of delay. In scheduling problems, lag time is an important consideration, as it can impact the overall efficiency and effectiveness of the scheduling solution. In specific scheduling scenarios, like the HFS scheduling problem involving lag time, this temporal gap becomes a critical parameter necessitating consideration for achieving the optimal scheduling solution (*Tran et al., 2023*).

After conducting an extensive literature review on the HFS problem, it is noteworthy that no existing studies have simultaneously considered unloading times, lag times, and transportation times for HFS scheduling problems. Consequently, this study aims to investigates the HFS scheduling problem with unloading times, lag times, and transportation times between stages. The problem consists of a serial set of stages, each containing identical parallel machines that process a given set of jobs. In this particular problem, each job undergoes processing on a machine within the first stage, followed by its unload (removal). Subsequently, the job is required to wait for a minimum duration of time, known as the lag time, before being transported to the subsequent stage. This is repeated until reaching the final stage where a machine carries out the processing of the job. Once the job is processed, it is removed from the machine and exits the system. Minimizing the maximum completion time or makespan is the objective function.

The investigation begins by defining the problem at hand. Moreover, an analysis of its complexity confirms its robust NP-hard categorization. One crucial aspect of the issue is pinpointed: its symmetric nature. This property signifies that scheduling from the initial stage to the final one (forward) or scheduling from the final stage to the initial one (symmetric) results in the same optimal solution. The significance of this attribute lies in the fact that all proposed algorithms are adapted to the symmetric problem, thereby broadening the scope for enhancing solution quality.

In addition a new set of lower bounds is proposed. These lower bounds can be divided into two types. In the first type, we relax the capacities of the stages, except for one, reducing the problem into a parallel machine scheduling problem with release dates and delivery times. In the second type, the lower bound is based on estimation of the minimum idle machine times.

In the third step, an existing heuristic (ADA) designed for addressing the parallel machine scheduling problem with release dates and delivery times (*Gharbi & Haouari, 2007*) is modified and repurposed into another heuristic (ADAU) to handle the parallel machine scheduling problem with release dates, unloading times, and delivery times.

Using the suggested heuristic (ADAU), we have formulated a two-phase heuristic for addressing the current problem under study. The initial phase involves constructing preliminary solutions, followed by a refinement process in the second phase. The latter heuristic employs the optimal solution for the parallel machine scheduling problem, considering release dates, unloading times, and delivery time, which is obtained through the ADAU heuristic. During the initial phase, we select an initial stage and allocate and solve a scheduling problem for parallel machines. Afterward, the process progresses to the subsequent stage, where another parallel machine scheduling problem is formulated and solved. This sequence is repeated until reaching the final stage. Next, a backward movement commences, beginning from the initially selected stage, towards the preceding stages. At each visited stage, a parallel machine problem is established and resolved. This process halts upon reaching the first stage. By concatenating all the schedules acquired from various stages, a feasible solution is achieved. By altering the starting stage, several feasible solutions are produced, and the best one is chosen as the initial solution.

The improvement phase focuses on enhancing the quality of the initial solution. This phase follows a similar procedure to the initial phase in terms of generating feasible solutions. Indeed, starting with the initial solution, we choose a starting stage where a parallel machine problem is defined and solved. If an enhancement is identified, it is passed on to the subsequent stages by updating the characteristics of the jobs. Following this, the next stage is visited, and a parallel machine scheduling problem is formulated and resolved. These iterations continue until reaching the final stage. Commencing from the final stage and moving backward to the preceding stage, where a parallel machine problem is formulated and resolved. This process is reiterated until reaching the initial stage. The forward and backward procedures continue until a predefined stopping condition is satisfied. By varying the starting stage, multiple improved solutions are generated, and the best one is retained.

The contributions of this study are as follows:

- **Novel investigation**: This is the first study, based on existing literature, to simultaneously investigate the HFS problem considering transportation, lag, and unloading times.
- **Identification of key properties**: The study identifies significant properties of the problem, particularly its symmetric nature. Utilizing this property helps improve the quality of the solutions.
- **Development of lower bounds**: New lower bounds are proposed, enabling the evaluation of solution quality through the relative gap. These lower bounds are derived from relaxations that simplify to polynomial parallel machine scheduling problems.
- **Heuristic algorithm presentation**: The study introduces an efficient two-phase heuristic algorithm capable of generating optimal or near-optimal solutions within a reasonable computational time frame. This heuristic is based on a novel algorithm designed to solve

parallel machine scheduling problems with release dates, unloading time, and delivery time.

- **Real-world application**: The research enables the modeling of real-world manufacturing systems that consider unloading, lag, and transportation times—factors often neglected in many industries. Practical applications include the steel industry and bioprocess industry.

The subsequent sections of this article are organized as follows. 'Literature review' formally defines the scheduling problem under study and presents some of its noteworthy properties. 'Proposed lower bounds' focuses on a new family of tight lower bounds. The proposed heuristic algorithm, consisting of two phases, is introduced in 'Two-phase heuristic solution'. 'Experimental results' entails a comprehensive experimental analysis that focuses on evaluating the effectiveness of the developed algorithms. Through this rigorous evaluation, the study provides valuable insights into the performance and efficacy of the proposed approaches in addressing the problem at hand. Finally, the article concludes by providing a summary of the main findings derived from the study. Additionally, it presents future research directions that requires further investigation and exploration in order to advance the field.

## LITERATURE REVIEW

The literature review of this study focuses on recent publications concerning the general HFS problem. Special attention is given to HFS scheduling problems incorporating transportation times, lag times, or unloading times. Ultimately, the study identifies the research gap.

### General HFS

The HFS has captured substantial interest within academic spheres and the manufacturing industry alike, thanks to its remarkable relevance and remarkable adaptability across a wide array of production systems. In make-to-order environments, there is a strong demand for flexible production systems, particularly because scheduling plans frequently face unforeseen disruptions. In the contemporary research landscape, there has been a discernible trend highlighting the efficacy of adaptable systems, such as the flexible flow shop, in adeptly managing the uncertainties characteristic of these environments (*Fattahi, Hosseini & Jolai, 2013*; *Gen, Gao & Lin, 2009*). Extensive literature exists regarding HFS, underscoring a substantial body of research and discourse within this references (*Lee & Loong, 2019*; *Ribas, Leisten & Framiñan, 2010*; *Ruiz & Vázquez-Rodríguez, 2010*; *Tosun, Marichelvam & Tosun, 2020*). These references provide comprehensive insights and detailed analyses of problems related to HFS. Below is a concise summary of recent publications that delve into different versions of the HFS.

*Liu et al. (2024)* examined the distributed HFS with blocking constraints. They proposed using greedy algorithms to tackle the problem, reducing idle machine time through active decoding techniques. Subsequently, a neighborhood search framework is implemented to increase the diversity of solutions. To generate effective initial solutions, they devised

a heuristic rule centered on blocking constraints. *Li et al. (2023a)* addressed an HFS problem that incorporates energy considerations. To address this challenging problem, a multi-objective Mixed-Integer Linear Programming (MILP) approach is proposed. Furthermore, the article introduces both a Q-learning technique and an enhanced genetic algorithm as solutions tailored specifically for this problem. The investigation by *Guan et al. (2023)* explores various solution representations for the HFS, emphasizing their respective advantages and limitations. It also deals with the task of striking a balance between narrowing down the solution space and maintaining an efficient search process within the confines of the limited solution space. *Liu et al. (2023)* concentrate on the dynamic HFS with re-entrant jobs, integrating factors such as skill levels and worker fatigue into their analysis. They utilize a multi-agent technique reinforced by deep learning to achieve a solution that approaches optimality. Thorough experimental studies illustrate the effectiveness of the algorithm put forth. In the study by *Ghodratnama, Amiri-Aref & Tavakkoli-Moghaddam (2023)*, researchers explore an MFFS problem that entails fuzzy maintenance time and robotic integration. They present a bi-objective mathematical model and utilize two multi-objective decision-making strategies: LP-metric and goal attainment (GA). The efficacy of these solution methods is evaluated and prioritized using the ''TOPSIS'' (Technique for Order of Preference by Similarity to Ideal Solution) methodology. In the study outlined in *Huang et al. (2023)*, diverse uncertain parameters linked to the production process in HFS are considered. To tackle the problem, they introduce a two-stage stochastic programming approach. To address this challenge effectively, they propose a novel iteration of the pointer-based discrete differential evolution (PDDE) algorithm, referred to as H-PDDE. This variant is crafted to enhance the performance and efficiency of the PDDE algorithm, particularly tailored for addressing the specified problem.

*Gholami & Sun (2023)*, address a distributed MFFS involving multiprocessor jobs. They frame the issue as a Markov Decision Process (MDP) and subsequently utilize a hybrid Q-learning-local search algorithm to resolve it. This approach combines the advantages of Q-learning, a reinforcement learning technique, with local search methods to effectively obtain near-optimal solutions for the given problem. In their work *Tran et al. (2023)*, improved mathematical integer formulations are introduced for the HFS, integrating chaining time-lag and time-varying resources. To reinforce these formulations, researchers devise valid inequalities. These inequalities are subjected to testing and evaluation, with the findings demonstrating their performance. In *Fernandez-Viagas, Molina-Pariente & Framinan (2018)*, a thorough investigation into constructive heuristics for the HFS is carried out. The assessed heuristics are put to experimental scrutiny and juxtaposed with four newly proposed counterparts. Two memory-centric constructive heuristics are unveiled, featuring a gradual integration of jobs into a partial sequence, while advantageous insertions are cataloged to form the eventual sequence. Furthermore, two constructive heuristics based on Johnson's algorithm are put forth as an alternative strategy. In their work *Li et al. (2023b)*, researchers addressed a challenge within the HFS related to batch processing in the sand casting industry. They introduced an upgraded cuckoo algorithm, integrating crossover and mutation operations to enhance its search capabilities. These enhancements

were aimed at replacing the traditional long and short flight strategies within the cuckoo algorithm.

In their study (*Wang et al., 2023*), researchers investigate a distributed two-stage HFS issue related to maintenance requirements. They introduce a mixed integer programming model and suggest employing a genetic algorithm to tackle this challenge. In their study (*Utama & Primayesti, 2022*), researchers delve into the HFS, incorporating energy considerations. They present an optimization algorithm that employs the Hybrid Aquila Optimizer (HAO) to tackle this specific issue. Within their study (*Shao, Shao & Pi, 2023*), researchers explore the distributed heterogeneous MFFS that integrates lot-streaming. They unveil a mixed-integer linear programming model and suggest a series of constructive heuristics, alongside an iterated local search algorithm. These constructive heuristics, grounded in time-based rules, form a notable part of their approach.

Scheduling multiprocessor tasks entails the necessity for a task to be processed simultaneously by multiple parallel processors (machines). Researchers have paid particular attention to scheduling multiprocessor tasks in hybrid flow shops (HFSMT) due to its significance in real-world applications. In this context authors in *Engin & Engin (2020)* addressed a HFSMT problem. This study proposes a novel memetic algorithm that integrates both global and local search methods to address the challenges of HFSMT scheduling problems. An intensive experimental study on benchmark test problems demonstrates that the proposed algorithms surpass existing ones in performance. Additionally, in *Engin & Engin (2018)* a HFSMT under the environment of a common time window is examined. In this research, a new memetic algorithm in which a global search algorithm is accompanied with the local search mechanism is developed to solve the HFSMT with jobs having a common time window. Memetic algorithm is tested using HFSMT problems. In *Kahraman et al. (2010)*, authors introduced a highly efficient parallel greedy algorithm for solving the HFSMT problem. Furthermore, they proposed four constructive heuristic methods to tackle HFSMT problems. Comparative computational results with previous works in the literature demonstrate the remarkable effectiveness of the proposed algorithms in terms of significantly reducing total completion time or makespan. *Engin, Ceran & Yilmaz (2011)* conducted a study that focused on an HFS scheduling problem involving multiprocessor tasks, where each job necessitates the use of more than one machine for processing. In their research, the authors proposed an efficient genetic algorithm specifically designed for tackling this problem.

## HFS with lag times

In a recent study conducted by *Tran et al. (2023)*, an improved and refined linear integer formulation is presented for the HFS. This advanced formulation takes into account time-varying resources and incorporates chaining time-lag constraints, thereby offering a more comprehensive approach to addressing the complexities of the problem. To enhance the robustness of the formulations, two valid inequalities are proposed. These inequalities serve to strengthen the mathematical models and improve their ability to accurately represent and solve the problem at hand. The revised formulations are evaluated through a benchmarking process, and the results indicate that the valid

inequalities significantly improve their performance. *Tran et al. (2021)* Introduces a novel mathematical formulation for the HFS problem, which considers time-varying resources and exact time-lag constraints. A comparison of this formulation to others is made, and it is found that it always guarantees a feasible solution for all instances. In a study conducted by *Harbaoui, Khalfallah & Bellenguez-Morineau (2018)*, a two-stage HFS problem incorporating precedence constraints, setup times, and lag times is examined and addressed. Due to the complexity of the studied problem, a hybrid genetic algorithm (HGA) is presented. The proposed procedures are successfully demonstrated to efficiently solve the problem under study within acceptable CPU time constraints. This outcome is the culmination of a rigorous experimental study. In a study conducted by *Javadian et al. (2012)*, the HFS problem is investigated, taking into account both setup and lag times. In Authors proposed a mathematical model that solves efficiently small sizes instances. In addition, for medium and large sizes problems a meta-heuristic algorithm based on the immune algorithm is proposed. The computational results indicate that the proposed algorithm can generate near-optimal solutions in a short period of time. In *Botta-Genoulaz (2000)*, authors studied the HFS problem under precedence constraints, lag times and due dates. In the latter research work, six new heuristics aiming to solve the problem are presented. An experimental study is performed to assess the performance and robustness of these algorithms, and the computational results substantiate the high quality of the obtained solutions.

## HFS with transportation times

The authors in *Gheisariha et al. (2021)* investigated the HFS problem associated with transportation time, sequence setup time, and rework. To solve the problem, they proposed an efficient Enhanced Harmony Search Algorithm. In *Lei et al. (2020)*, the HFS problem with dynamic transportation waiting time is examined, and a memetic algorithm is proposed to provide a near-optimal solution. In the work by *Naderi, Zandieh & Shirazi (2009b)*, the authors focus on addressing the HFS problem that considers both setup time and transportation time. They propose an effective solution approach based on the electromagnetism metaheuristic to solve this problem. In *Naderi et al. (2009a)*, the HFS problem with transportation and setup times is examined, with respective objective functions of total tardiness and total completion time. A proposed solution entails using a metaheuristic based on the simulated annealing algorithm in order to overcome this problem. Additionally, *Zabihzadeh & Rezaeian (2016)* conducted a study examining the HFS problem, with a specific focus on incorporating release dates and transportation time with robots. The objective of their study is to minimize the makespan in this context. Both ant colony optimization algorithm and genetic algorithms are proposed to solve this problem.

*Zhong & Lv (2014)* examined a two-stage HFS which considers the transportation times. In the latter problem, the machine-stage configuration involves two machines in the second stage and a single machine in the first stage. The transportation between stages is performed by a one-capacity transporter. To find a solution that is close to optimal, the researchers propose an efficient heuristic algorithm. *Zhu (2012)* addresses the HFS

problem with batching restrictions and transportation times. In a specific case, the author suggests a heuristic approach with polynomial time complexity to address the problem. Furthermore, for the general case, a heuristic with pseudo-polynomial complexity is developed as a solution. *Elmi & Topaloglu (2013)* examines the HFS problem with robots' transportation and blocking restrictions, proposing an efficient simulated annealing metaheuristic to minimize makespan. The presented algorithms are subjected to a comprehensive experimental evaluation. *Chikhi et al. (2015)* focus on examining the two-stage HFS problem that involves the inter-stage transportation with robots. In the considered problem, the initial stage comprises two dedicated machines operating in parallel, while the subsequent stage is composed of a single machine. A thorough investigation of the problem and its various specific cases is conducted, including those that are considered NP-hard. In response to the NP-hard variants, the researchers develop two efficient heuristics.

### HFS with unloading times

In their publication (*Gupta & Tunc, 1994*), the authors investigate the HFS problem that incorporates unloading times and setup times. The authors propose a set of heuristics, which are thoroughly evaluated through an experimental study. The results of the study show the efficiency and effectiveness of these algorithms in addressing the studied HFS problem. *Botta-Genoulaz (2000)* focuses on studying an HFS problem that considers both precedence constraints and unloading times. In order to provide a near-optimal solution for the addressed scheduling problem, the authors propose a set of six heuristic, which are demonstrated to be efficient through an extensive computational study. The authors of the study conducted by *Low (2005)* investigate an HFS problem that involves unloading and setup times. Notably, the parallel machines within each stage are unrelated, adding complexity to the problem. A heuristic algorithm is developed by the authors to obtain an initial feasible solution. A simulated annealing metaheuristic is then used to enhance this initial feasible solution. To assess the efficiency of the developed algorithms, an experimental study is conducted, providing substantial evidence of their effectiveness. A real-life HFS problem with unloading times and additional constraints is investigated in *Gicquel et al. (2012)*. A mixed integer linear program (MILP) is proposed. This MILP is then improved by including several new inequalities. The obtained algorithm is providing optimal solution for the industrial test problems.

### Research gap

Based on the current literature, the HFS scheduling problem involving lag times, transportation and unloading times has yet to be addressed. Therefore, there is a research gap that needs to be filled by studying this type of problem and developing effective procedures to solve it.

## PROBLEM STATEMENT

This section will provide a detailed explanation of the problem under study, along with a thorough examination of its key features.

## Definition of the studied problem

The HFS with lag time, transportation time, and unloading time (HFSULT) is defined as follows: A set of $n$ tasks, denoted $\{J = 1, 2, \ldots, n\}$, must be processed in a shop comprising a series of $K$ production stages, denoted $\{ST = S_1, S_2, \ldots, S_K\}$. Each stage, $S_i$ contains $m_i$ parallel machines which are identical, denoted $M_{i,1}, M_{i,2}, \ldots, M_{i,m_i}$ (i $= 1, 2, \ldots, K$). To be consider as HFSULT, at least one stage contains more than one machine. It is also assumed that $n$ is greater than the maximum of $m_i$ for all $i$ between 1 and $K$. The tasks must be processed consecutively, starting with stage $S_1$ and ending with stage $S_K$. Following the completion of processing in stage $S_i$, each task $j$ is unloaded from the machine and then subjected to a lag time (such as cooling, cleaning, or fermentation) during which no further treatment is carried out. Once the lag time has elapsed, task $j$ is transported to stage $S_{i+1} (1 \leq i \leq K-1)$. In the HFSULT problem, the processing time for task $j$ in stage $S_i$ is identical across all machines. Additionally, the machine responsible for processing a task $j$ remains unavailable until the task is entirely unloaded. The following notations are utilized to describe the problem:

- $pr_{i,j}$: the processing time of task $j$ at stage $S_i$.
- $un_{i,j}$: the unloading time of task $j$ at stage $S_i$.
- $lg_{i,j}$: the lag time of task $j$ at stage $S_i$.
- $tr_{i,j}$: the transportation time of task $j$ from stage $S_i$ to stage $S_{i+1} (1 \leq i \leq K-1)$.

Following the completion of task processing, it is crucial to note that the unloading operation may not commence immediately. The unloading and processing operations are completely separate, meaning that while a task is being unloaded or still present in the machine, the machine cannot be utilized for the processing of additional tasks. Additionally, it should be noted that during the lag time, neither transportation nor processing can be carried out.

The following assumptions serve as the basis for scheduling activities:

- No interruptions or preemption are allowed during processing.
- In a stage, each job is treated by only one machine.
- It is not possible for multiple jobs to be processed by the same machine simultaneously.
- Machines have the possibility to remain idle and await assignment for the next job.
- No breakdowns are assumed to occur with the machines.
- Between stages, buffers with an infinite capacity are assumed to be present.
- All jobs strictly follow a predefined route during processing. This route initiates from the first stage and progresses sequentially until reaching the final stage.
- There is no disparity in the processing time among the machines within each production stage since the machines are identical.
- $pr_{i,j}$, $un_{i,j}$, $lg_{i,j}$, and $tr_{i,j}$ (i $=1, \ldots, K$; j $=1, \ldots, n$) are deterministic and have positive integer values.
- All machines and jobs are assumed to be available starting from time zero.

Finding a schedule $\pi^*$, that optimized the makespan is the objective of this study. The makespan is defined as the maximum completion time and is represented by the following

**Table 1  Processing, unloading, and transportation times of example 1.**

| $j$ | 1 | 2 | 3 | 4 |
|---|---|---|---|---|
| $pr_{1,j}$ | 2 | 3 | 2 | 4 |
| $un_{1,j}$ | 3 | 3 | 2 | 3 |
| $lg_{1,j}$ | 2 | 3 | 2 | 2 |
| $tr_{1,j}$ | 3 | 3 | 2 | 2 |
| $pr_{2,j}$ | 2 | 2 | 2 | 2 |
| $un_{2,j}$ | 2 | 3 | 3 | 3 |
| $lg_{2,j}$ | 3 | 2 | 2 | 2 |
| $tr_{2,j}$ | 2 | 2 | 2 | 3 |
| $pr_{3,j}$ | 2 | 2 | 3 | 3 |
| $un_{3,j}$ | 3 | 2 | 2 | 2 |

expression:

$$C_{max}(\pi^*) = \max_{j \in J}(c_{K,j}(\pi^*)) \qquad (1)$$

Here, $c_{K,j}(\pi^*)$ is the exit time of job $j$ from the system (leaving the last stage $S_K$). In addition, $c_{K,j}(\pi^*) = CU_{K,j}(\pi^*) + lg_{K,j}(\pi^*)$ where $CU_{K,j}(\pi^*)$ is the finishing date of the unloading operation of job $j$ in $S_K$.

$FH_K, \left( \left( PM^{(l)} \right)_{l=1}^{K} \right) | un_{i,j}, lg_{i,j}, tr_{i,j} | C_{max}$ is the notation of the studied problem following the three-field notation (*Graham et al., 1979*).

Here, $FH_K$ represents the problem type, where "$FH$" indicates that the problem is a flow shop hybrid scheduling problem, and $K$ denotes the number of stages in the production process. The second part of the first field, $\left( \left( PM^{(l)} \right)_{l=1}^{K} \right)$, describes the set of machines available for processing in each stage. This field specifies the existence of parallel machines, represented as $PM^{(l)}$, within each stage denoted as $S_l$.

The second field, $| un_{i,j}, lg_{i,j}, tr_{i,j} |$, specifies the parameters of the problem, including unloading time $un_{i,j}$, lag time $lg_{i,j}$, and transportation time $tr_{i,j}$, for each job $j$ in each stage $S_i$. Finally, $C_{max}$ in the third field represents the objective function, which is to minimize the maximum completion time over all jobs.

The above points can be illustrated using the following example.

**Example 1:** Consider $n = 4, K = 3$, and $m_3 = m_2 = m_1 = 2$. Table 1 presents the processing times $pr_{i,j}$, unloading times $pr_{i,j}$, lag times $lg_{i,j}$, and transportation times $tr_{i,j}$ for each job $j$ in each stage $S_i$, $(i = 1, 2, .., K$ and $j \in J)$.

Figure 1 depicts a feasible scheduling solution with a makespan of $C_{max} = 34$ for the problem at hand, as per the example discussed earlier.

In Fig. 1, it can be observed that task 3 remains on machine $M_{1,1}$ even after its processing is completed at time 7. An unloading operation starts at time 10 and finishes at time 12. During this time window [7;10], the machine is not accessible and cannot be utilized for jobs treatment. This observation involves that the unloading operates independently of the required processing time.

Moreover, in the first stage, the lag time for task 1 starts once the unloading operation is completed, which occurs at time 5. The lag time period ends at time 7, but transportation

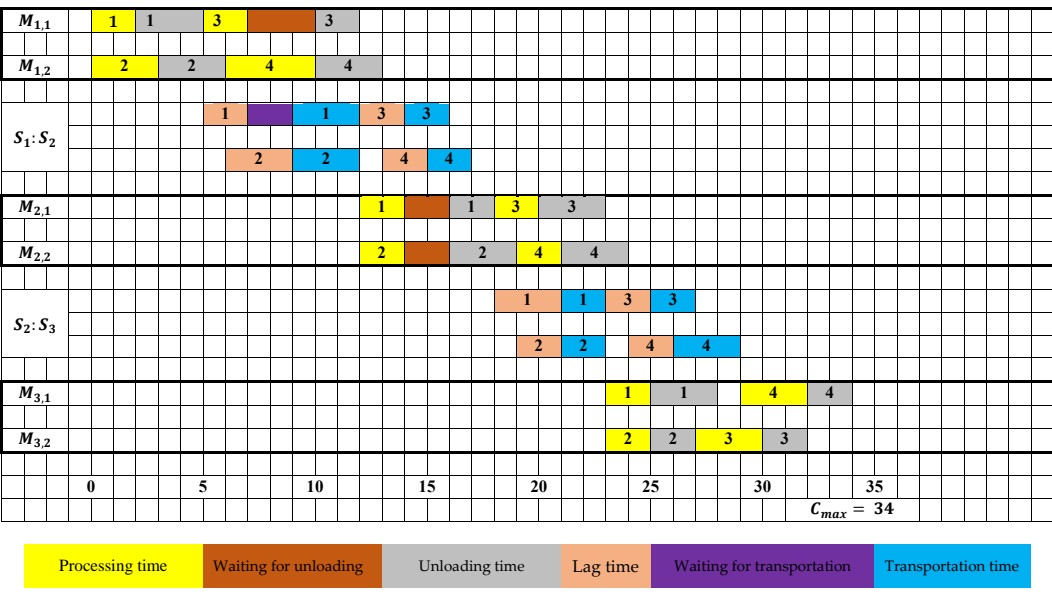

Processing time    Waiting for unloading    Unloading time    Lag time    Waiting for transportation    Transportation time

**Figure 1**   **Gantt chart of a feasible schedule for example 1.**

is postponed until time 9. Hence, it can be inferred that the lag time is independent of the transportation operation.

## Properties of the problem
### Time complexity of the problem
According to the following lemma (Lemma 2) the problem under study is complex for solving.

**Lemma 2:** *The problem under investigation is strongly NP-hard.*

**Proof.** *The particular case $K = 2$ and $un_{i,j} = lg_{i,j} = tr_{i,j} = 0$ defines the scheduling problem $H_2, \left(\left(PM^{(l)}\right)_{l=1}^{2}\right) || C_{\max}$. According to (*Gupta & Tunc, 1991*), $H_2, \left(\left(PM^{(l)}\right)_{l=1}^{2}\right) || C_{\max}$ is strongly NP-hard. Therefore, the general problem $FH_K, \left(\left(PM^{(l)}\right)_{l=1}^{K}\right) | un_{i,j}, lg_{i,j}, tr_{i,j} | C_{\max}$ is NP-hard.*

### The symmetric problem
This subsection is devoted to the introduction of the symmetric problem as well as its proprieties. Indeed, the study demonstrates that the optimal solutions for the original problem and its symmetric are equivalent and have the same makespan. As a result, to improve the solution quality, we conducted a systematic investigation of the symmetric problem.

**Definition 3:** *The symmetric problem (backward) of $FH_K, \left(\left(PM^{(l)}\right)_{l=1}^{K}\right) | un_{i,j}, lg_{i,j}, tr_{i,j} | C_{max}$, involves beginning scheduling from the last stage $S_K$ toward the first one $S_1$. By interchanging the roles of the stages, the symmetric (backward) problem can be derived, leading to a symmetric arrangement.*

According to the aforementioned definition (definition 3), the following notations are presented. The forward problem is starting scheduling in the first stage, $S_1$, and ending at the last stage, $S_K$.

● The symmetric problem is starting scheduling in the last stage, $S_K$, and ending at the first stage, $S_1$.

**Definition 4**

● The following notations represent the stages and machines respectively in the context of the symmetric problem: $m_k^B = m_{K-k+1}$, and $S_k^B = S_{K-k+1}$ (k =1 , …, K).

● The machines are represented by the following notations: $M_{k,l}^B = M_{K-k+1,l}$ (k =1 ,…, K; l =1 , 2,…, $m_k^B$).

● The processing, unloading, lag , and transportation times of the backward (symmetric) problem are denoted respectively as follows: $pr_{k,j}^B, un_{k,j}^B , lg_{k,j}^B$ , and $tr_{k,j}^B$.

● We have: $tr_{k,j}^B = lg_{K-k+1,j}, pr_{k,j}^B = un_{K-k+1,j}, lg_{k,j}^B = tr_{K-k+1,j}$, and $un_{k,j}^B = pr_{K-k+1,j}$.

● $FH_K, \left(\left(PM^{(l)}\right)_{l=1}^K\right)\left|un_{k,j}^B, lg_{k,j}^B, tr_{k,j}^B\right|C_{\max}$ is the three-field notation of the symmetric problem.

This following result highlights the significance of exploring the symmetric problem.

**Proposition 5:** Every feasible solution for the forward problem $FH_K, \left(\left(PM^{(l)}\right)_{l=1}^K\right)$ $\left|un_{i,j}, lg_{i,j}, tr_{i,j}\right|C_{max}$ can be converted into a feasible solution for the symmetric problem $FH_K, \left(\left(PM^{(l)}\right)_{l=1}^K\right)\left|un_{i,j}^B, lg_{i,j}^B, tr_{i,j}^B\right|C_{max}$.

Furthermore, the makespans for both schedules exhibit similarity.

***Proof***. A feasible solution $FS$ for the $FH_K, \left(\left(PM^{(l)}\right)_{l=1}^K\right)$ $\left|un_{i,j}, lg_{i,j}, tr_{i,j}\right|C_{max}$ problem can serve as the foundation for generating a feasible solution $FS^B$ for the $FH_K, \left(\left(PM^{(l)}\right)_{l=1}^K\right)\left|un_{i,j}^B, lg_{i,j}^B, tr_{i,j}^B\right|C_{max}$ problem. This can be accomplished by maintaining identical sequences and assignments on each machine. In the forward problem, the time scale is denoted by $''t''$, whereas in the backward problem, the time scale is represented by $t^B$, where $t^B = C_{max} - t$. The makespans of both feasible schedules $FS$ and $FS^B$ are identical since they share the same critical path. Similarly, utilizing the method outlined above, it becomes possible to convert a valid schedule obtained for the symmetric problem into an feasible schedule for the investigated problem.

In order to illustrate the concept of symmetry, Example 1 is revisited, and Table 2 presents the processing, unloading, lag, and transportation times associated with the symmetric problem.

Figure 2 displays a feasible solution to the symmetric problem, which is achieved by reversing the time scale from "$t$" to $(C_{max} - t)$.

**Table 2  A symmetric problem's processing, unloading, lag times, and transportation times.**

| $j$ | 1 | 2 | 3 | 4 |
|---|---|---|---|---|
| $pr_{1,j}^B$ | 3 | 2 | 2 | 2 |
| $un_{1,j}^B$ | 2 | 2 | 3 | 3 |
| $lg_{1,j}^B$ | 2 | 2 | 2 | 3 |
| $tr_{1,j}^B$ | 3 | 2 | 2 | 2 |
| $pr_{2,j}^B$ | 2 | 3 | 3 | 3 |
| $un_{2,j}^B$ | 2 | 2 | 2 | 2 |
| $lg_{2,j}^B$ | 3 | 3 | 2 | 2 |
| $tr_{2,j}^B$ | 2 | 3 | 2 | 2 |
| $pr_{3,j}^B$ | 3 | 3 | 2 | 3 |
| $un_{3,j}^B$ | 2 | 3 | 2 | 4 |

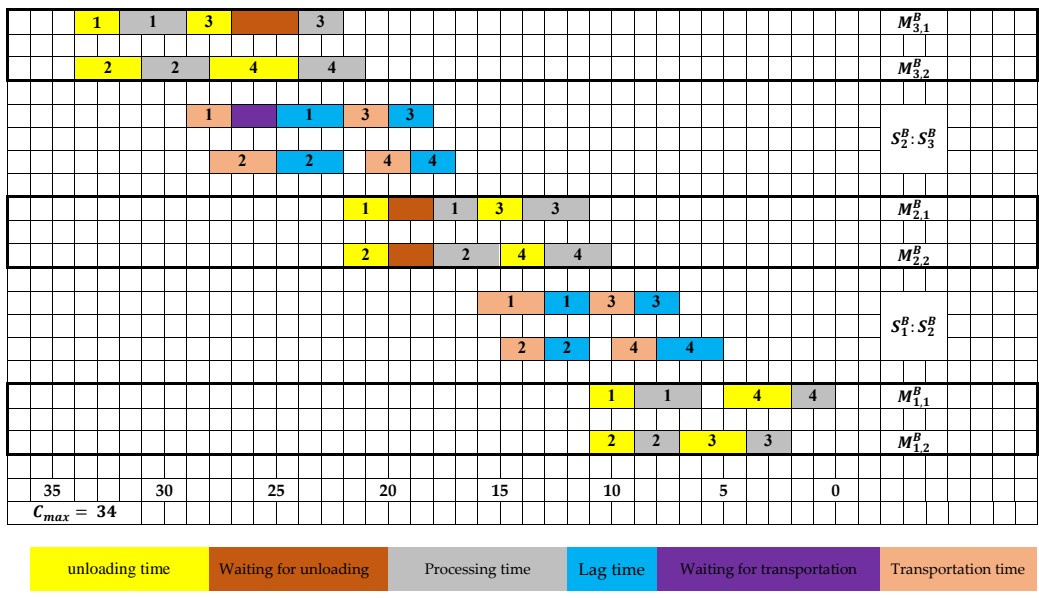

**Figure 2  Gantt chart of a feasible schedule for the symmetric problem relative to example 1.**

Proposition 5 entails the following significant corollary.

**Corollary 6:** *The* $FH_K, \left(\left(PM^{(l)}\right)_{l=1}^{K}\right) |un_{i,j}, lg_{i,j}, tr_{i,j}| C_{max}$ *problem and its symmetric counterpart,* $FH_K, \left(\left(PM^{(l)}\right)_{l=1}^{K}\right) \left|un_{i,j}^B, lg_{i,j}^B, tr_{i,j}^B\right| C_{max}$, *share the same optimal makespan.*

*Proof.* This outcome directly follows from Proposition 5.

The subsequent proposed procedures, including lower bounds and heuristic, are systematically employed to tackle the symmetric problem, resulting in enhanced outcomes. This observation directly follows from Corollary 6.

In the following sections, useful notations will be presented.

In each stage $S_k$ (k $=1$, $2$, $\ldots$, $K$) and for $j \in J$, $re_{k,j}$ given below is the release date.

$$\begin{cases} re_{k,j} = \sum_{i=1}^{k-1} \left( pr_{i,j} + un_{i,j} + lg_{i,j} + tr_{i,j} \right) \text{ if } k > 1 \\ re_{k,j} = 0 \text{ if } k = 1 \end{cases} \tag{2}$$

For $S_k$ (k $= 1$, $2$, $\ldots$, $K$) and for $j \in J$, $qe_{k,j}$ given below is the delivery time.

$$\begin{cases} qe_{k,j} = lg_{k,j} + tr_{k,j} + \sum_{i=k+1}^{K} \left( pr_{i,j} + un_{i,j} + lg_{i,j} + tr_{i,j} \right) \text{ if } k < K \\ qe_{k,j} = 0 \text{ if } k = K \end{cases} \tag{3}$$

Furthermore, for each stage $S_k$ (k $=1$, $\ldots$, $K$):

- $\overline{re_{k,j}}_{(i)}$ is the $i$th value in the increasingly sorted list of $re_{k,j}$'s ($j = 1, \ldots, n$).
- $\overline{qe_{k,j}}_{(i)}$ is the $i$th value in the increasingly sorted list of $qe_{k,j}$'s ($j = 1, \ldots, n$).
- $\overline{\left( un_{k,j} + pr_{k,j} \right)}_{(i)}$ is the $i$th value in the increasingly sorted list of $\left( un_{k,j} + pr_{k,j} \right)$'s ($j = 1, \ldots, n$).
- $\overline{\left( lg_{k,j} + tr_{k,j} \right)}_{(i)}$ is the $i$th value in the increasingly sorted list of $tr_{k,j}$'s ($j = 1, \ldots, n$).

# PROPOSED LOWER BOUNDS

In this section, we introduce lower bounds for the problem under investigation. Two categories of lower bounds are incorporated in this study. The first type comprises stages where capacity is relaxed for all but one element. The second type involves determining a lower bound by estimating the minimum idle time for each stage. The effectiveness of the heuristic is assessed by evaluating the relative gap between the upper and lower bounds. This is the objective of developing lower bounds.

## Lower bound based on capacity relaxation

Suppose that all stages, with the exception of one stage $S_k$ are equipped with an infinite number of machines. Once a task reaches a relaxed stage, it undergoes immediate processing. Consequently, the scheduling problem for stage $S_k$ can be represented as $P_m| \ re_{k,j}, un_{k,j}, qe_{k,j}|C_{\max}$, where the problem characteristics encompass:1) the number of machines ($m = m_k$), 2) release dates ($re_{k,j}$), 3) unloading times ($un_{k,j}$), and 4) delivery times ($qe_{k,j}$).

To introduce further relaxation to this problem, one can exclude the idle time after the completion of task processing and the commencement of the unloading for each job. By applying this relaxation, we obtain a $P_m| \ re_{k,j}, qe_{k,j}|C_{\max}$ problem. In this scenario, each task $j$ is assigned a processing time ($pr_{k,j} + un_{k,j}$). In their study (*Carlier, 1987*) of parallel machine scheduling with release and delivery ($P_m| \ r_j, q_j|C_{\max}$), the researchers introduced a new lower bound. By using the aforementioned lower bound as a reference, we can derive the following lower bound specific to stage $S_k$.

$$LB_k^S = \left\lceil \frac{1}{m_k} \left( \sum_{i=1}^{m_k} \overline{re_{k,j}}_{(i)} + \sum_{j=1}^{n} pr_{k,j} + \sum_{j=1}^{n} un_{k,j} + \sum_{i=1}^{m_k} \overline{qe_{k,j}}_{(i)} \right) \right\rceil. \tag{4}$$

***Proof***: For the problem $P_m|r_j, q_j|C_{\max}$, *Carlier (1987)* established that

$$LB = \left\lceil \frac{1}{m} \left( \sum_{i=1}^{m} \overline{r}_{j(i)} + \sum_{j=1}^{n} p_j + \sum_{i=1}^{m} \overline{q}_{j(i)} \right) \right\rceil$$ is a lower bound. Here, $\overline{r}_{j(i)}$ and $\overline{q}_{j(i)}$ represent

the $i$th release date and delivery time, respectively, sorted in increasing order. In stage $S_k$, let $m = m_k$, $r_j = re_{k,j}$, and $q_j = qe_{k,j}$. By omitting the idle time between the completion of a job's processing and the beginning of its unloading operation (as a relaxation), we define $p_j = pr_{k,j} + un_{k,j}$ Therefore, Therefore, $\overline{r}_{j(i)} = \overline{re_{k,j}}_{(i)}$, $\overline{q}_{(i)} = \overline{qe_{k,j}}_{(i)}$. Thus, the expression given in Eq. (4) indeed represents a lower bound for the studied problem.

It is important to note that Eq. (4) does not explicitly account for the transportation and lag times. However, these two parameters are implicitly considered and incorporated within $\overline{re_{k,j}}_{(i)}$ and $\overline{qe_{k,j}}_{(i)}$.

Therefore, we can conclude that the following result can be derived.

**Proposition 7:** The following expression represents a lower bound with complexity $O(Kn)$.

$$LB^S = \max_{1 \le k \le K} \{LB_k^S\}. \tag{5}$$

***Proof***: Eq. (4) involves that $LB_k^S$ is a lower bound for $P_{m_k}|re_{k,j}, qe_{k,j}|C_{\max}$ at stage $S_k$. By exploring all stages, the expression $LB^S = \max_{1 \le k \le K} \{LB_k^S\}$ yield a valid lower bound for the studied problem. The primary computational effort involved in calculating $LB_k^S$ is the sorting of $\overline{re_{k,j}}_{(i)}$ and $\overline{qe_{k,j}}_{(i)}$, which has a time complexity of $O(n)$. Therefore, by exploring all the $K$ stages, the time complexity of $LB^S$ is $O(Kn)$.

## An idle time based lower bound

This subsection aims to derive a secondary lower bound by introducing relaxation to a particular parallel machine scheduling problem. The purpose of this relaxation is to evaluate the minimum idle time within stage $S_k$. Specifically, we consider stage $S_{k-1}$ (if $k \ge 2$) and stage $S_{k+1}$ if (if $k < K$). The scheduling problem under consideration is a parallel machine problem with release dates, with the objective of minimizing the sum of completion times. The notation of the above mentioned problem is $P_m|r_j|\sum C_j$. This problem is defined as follows. A set of $m$ identical parallel machine has to process without preemption $n$ jobs. Each job $j$ is subject to a release date constraint (mentioned by $r_j$ in the second field of the previous three-field notation). Each job $j$ is processed during $p_j = pr_{k,j} + un_{k,j}$ units of time in a machine. The primary objective of this problem is to identify a viable schedule that satisfies the specified constraints while minimizing the total completion time, represented by the sum of all task completion times $\sum C_j$. According to *Yalaoui & Chu (2006)*, this problem is recognized as NP-hard.

$P_m|r_j|\sum C_j$ can be relaxed by permitting the job splitting. In this relaxed problem, jobs are permitted to be interrupted and resumed at any time, and they can also be executed concurrently on multiple machines. According to *Yalaoui & Chu (2006)*, the Shortest Remaining Processing Time (SRPT) algorithm is capable of effectively solving this relaxation by employing job splitting, with a time complexity of $O(n\log(n))$. In simpler terms, the SRPT rule selects the job $j$ with the shortest remaining processing time at any

given time $t$. Following that, the available machines schedule portions of job $j$ until either the entire job is processed or another available job with a shorter remaining processing time is found. For the problem $P_m|r_j|\sum C_j$ a lower bound is obtained by summing the $n$ smallest completion times obtained by using the SRPT rule.

The sum of the $l$ smallest completion times, obtained through the SRPT rule (splitting relaxation) within stage $S_k$, is denoted as $JSRPM_k(l)$.

The second lower bound is presented below over Proposition 8.

**Proposition 8:** Let $LB_{2S}(k)$ expressed as follows.

$$LB_{2S}(k) = \lceil \frac{1}{m_k}(JSRPM_{k-1}(m_k) + \sum_{i=1}^{m_k}\overline{(lg_{k-1,j}+tr_{k-1,j})}_{(i)} + \sum_{j=1}^{n}(pr_{k,j}+un_{k,j})+$$

$$\sum_{i=1}^{m_k}\overline{(lg_{k,j}+tr_{k,j})}_{(i)} + JSRPM_{k+1}(m_k))\rceil. \tag{6}$$

Therefore, for the problem under study, $LB_{2S}(k)$ is a lower bound specific to stage $S_k$ ($2 \leq k \leq K$). Furthermore, the time complexity of $LB_{2S}(k)$ is $O(n\log n)$.

***Proof***: When analyzing an optimal schedule with an optimal solution of $C^*_{max}$ and focusing on a specific stage $S_k$ (where $2 \leq k \leq K$), the following notations are utilized:

- $f_i$ denotes the first task processed by the machine $M_{k,i}$.
- $l_i$ represents the last task that was unloaded from the machine $M_{k,i}$.
- $s_{k,f_i}$ signifies the start time of processing task $f_i$ on machine $M_{k,i}$.
- $Pr_{k,i}$: This represents the total processing time on machine $M_{k,i}$
- $UN_{k,i}$: This signifies the total unloading time on machine $M_{k,i}$
- $Id_{k,i}$: denotes the total idle time in machine $M_{k,i}$.

Clearly, we have:

$$s_{k,f_i}+Pr_{k,i}+Id_{k,i}+UN_{k,i}+qe_{k,l_i} \leq C^*_{max}. \tag{7}$$

Therefore,

$$\sum_{i=1}^{m_k}s_{k,f_i}+\sum_{i=1}^{m_k}Pr_{k,i}+\sum_{i=1}^{m_k}Id_{k,i}+\sum_{i=1}^{m_k}UN_{k,i}+\sum_{i=1}^{m_k}qe_{k,l_i} \leq m_k C^*_{max}, \tag{8}$$

In addition,

$$\sum_{i=1}^{m_k}Pr_{k,i}=\sum_{j=1}^{n}p_{k,j} \text{and} \sum_{i=1}^{m_k}UN_{k,i}=\sum_{j=1}^{n}un_{k,j} \tag{9}$$

Furthermore,

$$\sum_{i=1}^{m_k}\overline{qe_{k,j}}_{(i)} \leq \sum_{i=1}^{m_k}qe_{k,l_i} \tag{10}$$

In stage $S_{k-1}$, we have a parallel machine problem $P_{m_{k-1}}|re_{k-1,j}|\sum C_j$ and,

$$JSRPM_{k-1}(m_k)+\sum_{i=1}^{m_k}\overline{(lg_{k-1,j}+tr_{k-1,j})}_{(i)} \leq \sum_{i=1}^{m_k}s_{k,f_i}. \tag{11}$$

Based on Eqs. (8), (9), (10) and (11), the following relationship can be established:

$$\left\lceil \frac{1}{m_k}\left( JSRPM_{k-1}(m_k) + \sum_{i=1}^{m_k}\overline{tr_{k-1,j}}_{(i)} + \sum_{j=1}^{n}\left(pr_{k,j}+un_{k,j}\right) + \sum_{i=1}^{m_k}\overline{qe_{k,j}}_{(i)} \right)\right\rceil \leq C_{\max}^{*} \qquad (12)$$

By considering the symmetric problem and directing attention to stage $S_{k+1}$, one can illustrate the following:

$$JSRPM_{k+1}(m_k) + \sum_{i=1}^{m_k}\overline{(lg_{k,j}+tr_{k,j})}_{(i)} \leq \sum_{i=1}^{m_k}\overline{qe_{k,j}}_{(i)} \qquad (13)$$

This ends the first part of the proof.

The main effort computing $LB_{2S}(k)$, lies in determining $JSRPM_{k-1}(m_k)$ and $JSRPM_{k+1}(m_k)$ $(1 \leq k \leq K)$, which takes $O(n\log n)$ time.

The following corollary is presented as a direct consequence of the preceding proposition (Proposition 8).

**Corollary 9:** The expression for a valid lower bound on the problem can be formulated as follows:

$$LB_{2S} = \max_{1 \leq k \leq K}\{LB_{2S}(k)\} \qquad (14)$$

Additionally, the time complexity of $LB_{2S}$ is $O(Kn\log n)$.

***Proof***: The lower bound $LB_{2S}(k)$ is derived from the preceding proposition (Proposition 8). By taking the maximum value among all $LB_{2S}(k)$ for $1 \leq k \leq K$, denoted as $LB_{2S} = \max_{1 \leq k \leq K}\{LB_{2S}(k)\}$, we obtain a lower bound that has a time complexity of $O(Kn\log(n))$.

### A general lower bound

By jointly considering $LB_S$ and $LB_{2S}$, the lower bounds can be enhanced to be more comprehensive and robust. The following expression represents this strengthened lower bound:

**Corollary 13:** A lower bound for the examined problem is expressed as follows.

$$LB = \max\left(LB_{2S}, LB^{S}\right) \qquad (15)$$

***Proof***. Obvious.

## TWO-PHASE HEURISTIC SOLUTION

By using this two-phase heuristic, a high-quality near-optimal solution can be achieved for the $FH_K, \left(\left(PM^{(l)}\right)_{l=1}^{K}\right)\left|un_{i,j}, lg_{i,j}, tr_{i,j}\right|C_{\max}$ problem. The heuristic consists of two distinct phases, namely Phase 1 (P1) and Phase 2 (P2). In Phase 1, an initial schedule is generated by solving the $P_m|r_j, un_j, q_j|C_{\max}$ problem successively in each stage. Subsequently, in Phase 2, the initial solution is refined further. This phase also requires solving parallel machine scheduling problems, but in addition to the factors mentioned earlier, it takes into account a related variant $(P_m|r_j, un_j|L_{\max})$ that aims to minimize the maximum

lateness. By combining these two phases and considering the specific scheduling problem constraints, an effective and high-quality solution can be obtained for the studied problem.

In the current research work, the heuristic algorithm employed is the Approximate Decomposition Algorithm with Unloading (ADAU). This algorithm is tailored to generate a near-optimal solution for the NP-hard problem $P_m|r_j, un_j, q_j|C_{\max}$. ADAU builds upon the foundations of ADA heuristic (*Gharbi & Haouari, 2007*) and incorporates additional considerations for unloading times. Recall that ADA was designed to approximate solutions for the scheduling problem $P_m|r_j, q_j|C_{\max}$.

Taking into account the unloading, lag, and transportation times forms a crucial aspect of the ADAU algorithm. During each iteration of ADAU, the algorithm identifies the machines with the shortest and longest completion times. Subsequently, a parallel machine problem involving these two machines and the scheduled jobs is solved. In the case of multiple machines having the same completion time, ADAU employs random selection to ensure fairness and prevent bias in the selection process. These iterations persist until a predefined stopping criterion is met. Through our experiments, it has been observed that ADAU is a fast algorithm that generally produces optimal schedules. The following sections offer a comprehensive explanation of both phases (P1 and P2) in the two-phase heuristic algorithm.

## Initial feasible solution (Phase 1)

A constructive procedure is employed to generate a feasible schedule $\gamma_i$ for each starting stage $S_i (1 \leq i \leq K)$. The pseudocode for stage $S_i$ is presented in Algorithm 1 as follows:

---

**Algorithm 1: Initial feasible schedule $\gamma_i$ construction**

---

**Step 1:** Set $q_j = qe_{i,j}$, $r_j = re_{i,j}$, $un_j = un_{i,j}$, and $p_j = pr_{i,j}$ ($j \in J$)

**Step 2:** *Using ADAU and solve $P_{m_i}|r_j, un_j, q_j|C_{\max}$ (**Step 1**).*
*Set the finishing unloading date as $CU_{i,j}$ ($j \in J$).*
*If $i == K$ Then Go to* **Step 4**.

**Step 3:** *For $s = i + 1$ to $K$*
**Step 3.1:** *Set $p_j = pr_{s,j}$, $un_j = un_{i,j}$, $r_j = CU_{s-1,j} + lg_{s-1,j} + tr_{s-1,j}$, and $q_j = qe_{s,j}$ ($j \in J$).*
**Step 3.2:** *Solve $P_{m_s}|r_j, un_j, q_j|C_{\max}$ (obtained in Step 3.1) utilizing ADAU. Set the finishing unloading date as $CU_{h,j}$ ($j \in J$).*
*End (For)*

**Step 4:** *Set $UB_i = \max_{j \in J}(CU_{K,j} + lg_{K,j})$. If $i == 1$ Then Go to* **Step 6**.

**Step 5:** *For $s = i - 1$ to $1$*
**Step 5.1:** *Set $T_{s+1,j}$ the beginning processing of $j$ in $S_{s+1}$*
$d_j = T_{s+1,j}$ $un_j = un_{i,j}$ $p_j = pr_{s,j}$, and $r_j = re_{s,j}$ ($j \in J$).
**Step 5.2:** *Solve $P_{m_s}|r_j, un_j|L_{\max}$ (defined in* **Step 5.1***) by using ADAU.*
**Step 5.3:** *In stage $S_l$ ($s + 1 \leq l \leq K$ Set $T_{l,j} := T_{l,j} + L^s_{\max}$ ($j \in J$).*
*Set $UB_i := UB_i + L^s_{\max}$.*
*End (For)*

**Step 6:** *Save $\gamma_i$ (schedule) and $UB_i$ (its makespan).*

---

The first step involves solving the problem $P_{m_i}|r_j, un_j, q_j|C_{\max}$ which is specific to stage $S_i$. This task is accomplished in Steps 1–2 by applying the ADAU heuristic. Moving on to the subsequent stage, $S_{i+1}$, the durations $CU_{i,j} + lg_{i,j} + tr_{i,j}$ for each task $j \in J$ are designated as

the release dates $r_j$, where $CU_{i,j}$ represents the completion unloading dates. Subsequently, the scheduling problem identified as $P_{m_{i+1}}|r_j, un_j, q_j|C_{max}$, as defined in Step 3.1, is tackled and resolved. This is achieved by utilizing the ADAU heuristic once again in Step 3.2. This process is repeated iteratively, starting from stage $S_{i+2}$ and continuing until stage $S_K$, with Steps 3.1–3.2 being executed in the same sequential order.

To achieve a comprehensive solution, iterative scheduling is utilized for the upstream stages $S_{i-1}, \ldots, S_1$. More precisely, in stage $S_{i-1}$, a problem $P|r_j, un_j|L_{max}$ is formulated by assigning due dates $d_j$ to each task $j \in J$, where the due dates are set equal to the starting time in stage $S_{i-1}$ (Step 5.1). Subsequently, in Step 5.2, ADAU is employed to generate a schedule with a maximum lateness $L_{max}^{i-1}$. During this phase, two cases need to be addressed:

- Case 1: If the maximum lateness in Step 5.2 is greater than zero ($L_{max}^{i-1} > 0$), it is required to shift the starting time in the subsequent stages ($S_i, \ldots, S_K$) to the right by $L_{max}^{i-1}$ units of time.
- Case 2: Conversely, if the maximum lateness is less than or equal to zero ($L_{max}^{i-1} \le 0$), the schedules in the subsequent stages ($S_i, \ldots, S_K$) should be left-shifted by $L_{max}^{i-1}$ units of time.

By implementing these adjustments, it ensures that the tasks are completed within their assigned due dates and maintains the overall feasibility of the solution.

Both in Case 1 and Case 2, it is essential to update the upper bound $UB_i$ in Step 5.3 to reflect the adjusted starting times. The latter procedure is applied in each stage $S_{i-1}, \ldots, S_1$. As a result, a feasible schedule $\gamma_i$ is generated for the problem under investigation, ensuring that all the given constraints are satisfied. By modifying the starting stage $S_i$ ($1 \le i \le K$), $K$ schedules $\gamma_1, \ldots, \gamma_K$ are obtained. From these schedules, the one with the minimum makespan $UB$ is selected as the initial solution. By adopting this approach, the overall solution is guaranteed to achieve the minimum possible makespan while still being feasible within the given constraints of the problem.

To simplify the presentation of the various steps in the above procedure (Phase 1), a straightforward example comprising five stages is provided. The entire process is illustrated in the self-contained Fig. 3.

In this example, stage 4 is chosen as the starting point ($i = 4$). A parallel machine problem is set up and solved. The same procedure is repeated for the subsequent stage (stage 5) as part of the forward process. For each iteration, a parallel machine problem is defined and solved. Once the last stage is reached, a backward procedure starts from stage 3 ($i - 1 = 3$), and continues with stages 2 and 1. For each of these stages, a parallel machine problem is defined and solved. By reaching the first stage, feasible solutions for all five stages are obtained. These solutions are not conflicting, and their combination provides a feasible solution for the studied problem. The ADAU heuristic is used to solve each parallel machine scheduling problem. If the starting stage is 1, 2, 3, or 5, four other feasible solutions are obtained, and the best one is kept as the initial feasible solution. Additionally, the parameters of the parallel machine problems are updated according to Step 3.1 and Step 5.3.

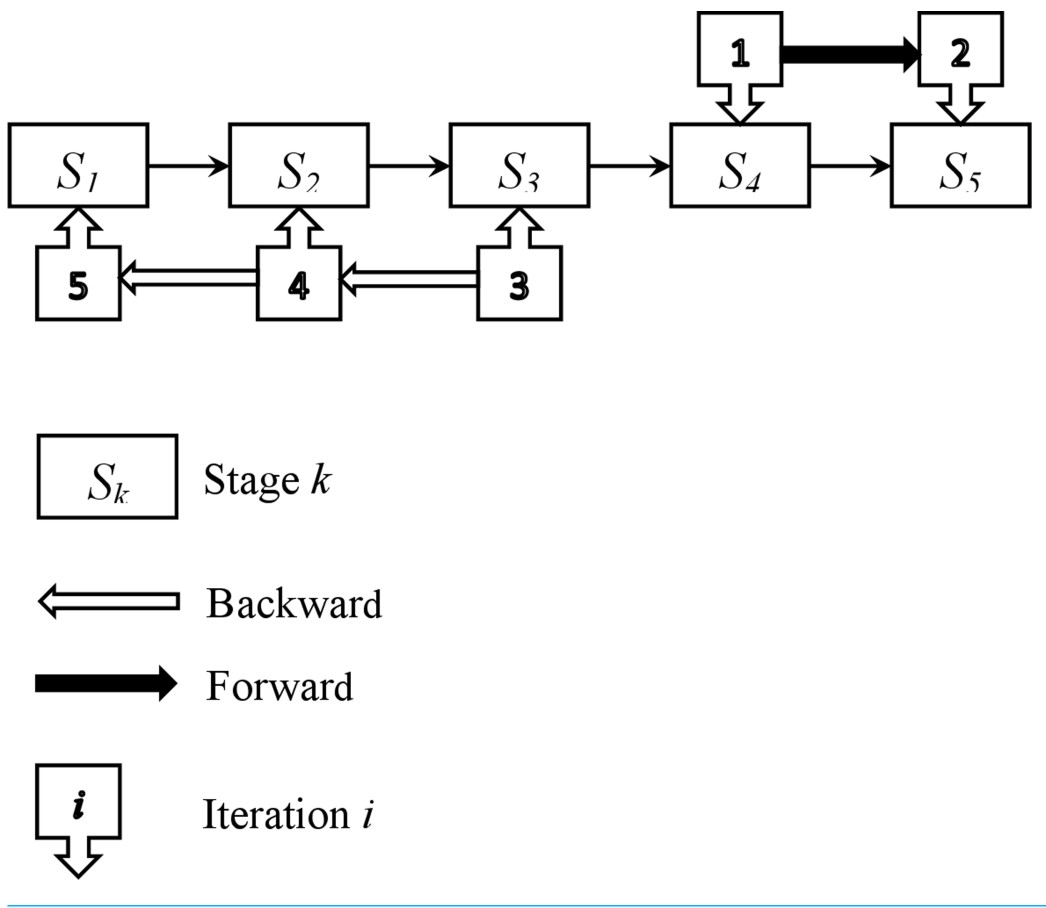

**Figure 3** Illustration of the initial feasible solution construction.

## The enhancement phase (Phase 2)

The iterative improvement phase begins after obtaining the feasible schedule $\gamma$ from Phase 1. A stopping criterion is set to determine when to terminate the improvement phase. In this phase, iterative refinements are applied to the schedules of all stages, excluding one specific stage $(S_h)$, in order to enhance the solution. To facilitate the rescheduling process, it is crucial to address the problem denoted as $P_{m_h}|r_j, un_j|L_{\max}$ as a fundamental aspect of the iterative adjustments. In the aforementioned problem, the release dates are defined as $r_j = CU_{h-1,j}$, where $CU_{h-1,j}$ represents the completion unloading time of job $j$ in the previous stage $(S_{h+1})$. Additionally, the due dates $d_j$ are determined based on the scheduled start time $T_{h+1,j}$ of the jobs in the next stage $(S_{h+1})$, denoted as $d_j = T_{h+1,j}$. In the last stage $S_K$, the due date $d_j$ is assigned as the makespan $UB$ ($d_j = UB$). After solving the rescheduling problem, two possible scenarios arise with respect to ADAU:

1. When $L_{\max}^h$ is negative (*i.e.*, $L_{\max}^h < 0$), it indicates that an improved solution has been achieved at stage $S_h$, with a makespan reduced by $\left|L_{\max}^h\right|$ units of time. In this case, a new schedule is created by shifting all tasks in stages from $S_h$ to $S_K$ to the left by $\left|L_{\max}^h\right|$ units of time. This adjustment guarantees that tasks are completed within their designated due dates and ensures the overall feasibility of the solution.

2. When $L_{\max}^h$ is greater than or equal to zero (*i.e.*, $L_{\max}^h \geq 0$), it indicates that no improvement has been achieved in this case. The iterative process is repeated for the subsequent stages until the convergence criterion is satisfied.

*Remark 14: The optimal solution $L_{max}^h$ of $P_{m_h}|r_j, un_j|L_{max}$ guarantees feasibility by satisfying $L_{max}^h \leq 0$. This condition is essential to guarantee that tasks are accomplished within their designated due dates, ensuring an optimal overall solution that adheres to the constraints of the problem.*

Subsequently, we provide a pseudo-code (Algorithm 2) that outlines the sequential steps of the improvement phase when initiating from stage $S_h$.

---

**Algorithm 2: Phase 2 (Enhancement phase)**

---

**Step 0:** *Initialization, Set $k = h$, $u = 0$.*

**Step 1:** *Set $u := u + 1$,*
*Set $p_j = p_{k,j}, r_j = CU_{k-1,j}$ $d_j = T_{k+1,j}$, and $un_j = un_{k,j}$ ($j \in J$).*

**Step 2:** *Solve $P_{m_k}|r_j, un_j|L_{\max}$ (defined in Step 1) by using ADAU.*

**Step 3:** **Step 3.1:** & *If $L_{\max}^k < 0$, Set $UB := UB + L_{\max}^k$ (enhanced solution)*
*Set $=$ 0.*

**Step 3.2:** *If $k == 1$ Then Go to* **Step 5**.

**Step 4:** *Set $k := k - 1$, If $u \leq 2K - 1$ Then Go to* **Step 1**, *Otherwise go to* **Step 9**.

**Step 5:** *Set $u := u + 1$*
*Set $r_j = CU_{k-1,j}, d_j = T_{k+1,j}, un_j = un_{k,j}$, and $p_j = p_{k,j}$ ($d_j = UB$ if $k = K$).*

**Step 6:** *Solve $P_{m_k}|r_j, un_j|L_{\max}$ (defined in* **Step 5**) *by using ADAU.*

**Step 7:** **Step 7.1:** *If $L_{\max}^k < 0$, Set $UB := UB + L_{\max}^k$ (an improvement is detected), Set $u = 0$.*

**Step 7.2:** *If $k == K$ Then Go to* **Step 4**.

**Step 8:** *Set $k := k + 1$, If $u \leq 2K - 1$ Then Go to* **Step 5**.

**Step 9:** *Save $\gamma^h$ (obtained schedule) and $UB$ (makespan).*

---

To commence the improvement phase, we begin by selecting an initial stage $S_h$ for starting the procedure, where a problem $P_{m_h}|r_j, un_j|L_{\max}$ is defined (Step 1). In Step 2, the problem $P_{m_h}|r_j, un_j|L_{\max}$ is solved by applying the ADAU heuristic. If improvements are identified, Step 3 involves performing a rescheduling of stage $S_h$. Similarly, if enhancements are identified in Step 4, the upstream stages $S_{h-1}$ $S_{h-2}, \ldots, S_1$ are subjected to rescheduling. When the heuristic procedure reaches the first stage in Step 3.2, the improvement phase is iteratively applied to each downstream stage $S_2, S_3, \ldots, S_K$ in Steps 5–8. During this process, Step 7.2 involves revisiting and rescheduling the upstream stages. The stopping criterion for the second phase is met when 2K-2 consecutive problems are solved without any improvement. The latter halting condition is represented by Step 9. This guarantees that the algorithm terminates either after a satisfactory number of iterations without any improvement or when an optimal solution is achieved.

The improvement phase operates on the feasible schedule $\gamma$ derived from Phase 1 as its input. The proposed improvement phase can commence with any starting stage $S_h$ ($h = 1, \ldots, K$), and $K$ improvement phases are carried out, each initiated with a different starting stage $S_h$ ($h = 1, \ldots, K$). As a result, $K$ feasible schedules are generated, and the

best schedule $\gamma^*$ is chosen from among them. By exploring various starting stages, the algorithm can discover a diverse range of solutions and ultimately select the one with the lowest makespan that satisfies all the problem's constraints. Incorporating multiple starting stages ensures the algorithm's robustness and its ability to handle diverse scenarios and input data effectively.

In the following, an example is provided to clearly explain the enhancement phase. This example involves five stages and assumes that an initial feasible solution has been obtained from phase 1. The entire process of this phase is illustrated in Fig. 4.

Stage 4 is chosen as the starting point ($h = 4$), where a parallel machine problem is set up and solved. This procedure is then repeated for the subsequent stage (stage 5) as part of the forward process. At each iteration, a parallel machine problem is defined and solved. Once the final stage (stage 5) is reached, a backward procedure begins from stage 5 and proceeds through stages 4, 3, 2, and 1 in that order. For each of these stages, a parallel machine problem is defined and solved. After reaching the first stage, a forward procedure starts again from stage 1 and continues to stage 5, with a parallel machine problem defined and solved at each stage. The backward and forward procedures are repeated until a stopping condition is met. These solutions are non-conflicting, and their combination provides a feasible solution for the problem under study. All parallel machine scheduling problems are solved using the ADAU heuristic. If the starting stage is 1, 2, 3, or 5, four other feasible solutions are generated, and the best one is retained as the improved solution.

**Remark 15:** *By exploring the symmetric problem the algorithm can effectively explore a wider search space, enabling the identification of better solutions. Through the exploitation of problem symmetry, the algorithm can significantly reduce redundancy and eliminate duplicates within the search space. This optimization facilitates faster convergence and enhances the quality of the obtained solutions. In summary, incorporating the symmetric problem within the two-phase heuristic enhances the overall effectiveness and efficiency of the algorithm, enabling it to discover optimal or near-optimal solutions to the HFSULT problem more effectively.*

## EXPERIMENTAL RESULTS

### Input instances

The test problems utilized in this study follow a similar generation approach as those in *Vandevelde et al. (2005)*. More specifically, $K \in \{2, 4, 6, 8, 10\}$ and $n \in \{10, 20, 40, 80\}$.

Table 3 presents the stage-machine configurations utilized in this study. It provides a comprehensive overview and details of the specific stage-machine combinations employed for the experimental analysis.

A noteworthy observation regarding the testbed is the presence of distinct patterns in the instances, stemming from their random distribution. These patterns can be characterized as follows:

- All-equal patterns: as 2-2-2-2-2-2-2-2-2-2.
- Increasing patterns: as 1-1-2-2-3-3-4-4-5-5.

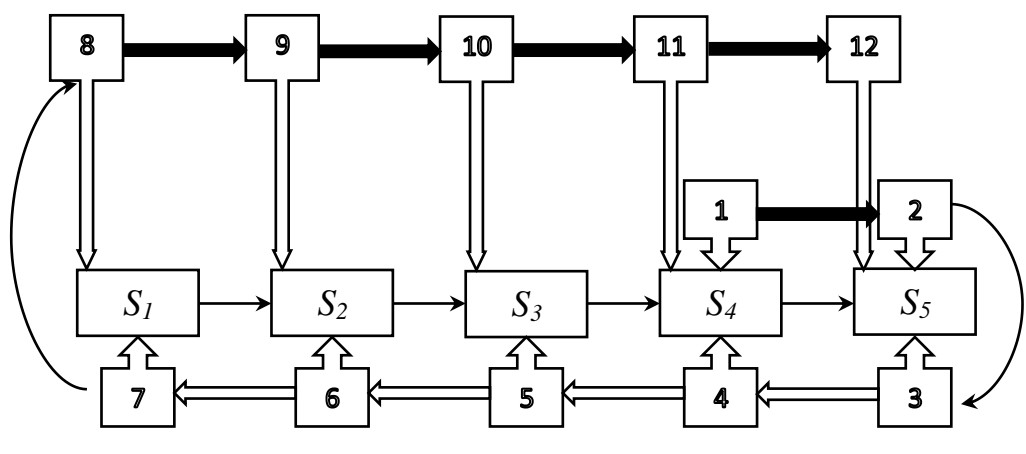

**Figure 4** Illustration of the improvement phase.

**Table 3** The used stages and machines configurations.

| Configuration | 2 stages | 4 stages | 6 stages | 8 stages | 10 stages |
|---|---|---|---|---|---|
| 1 | 2-2 | 2-2-2-2 | 2-2-2-2-2-2 | 2-2-2-2-2-2-2-2 | 2-2-2-2-2-2-2-2-2-2 |
| 2 | 1-2 | 2-4-4-6 | 1-2-3-4-5-6 | 1-1-2-2-3-3-4-4 | 1-1-2-2-3-3-4-4-5-5 |
| 3 | 1-4 | 2-4-2-4 | 1-2-3-1-2-3 | 1-3-1-3-1-3-1-3- | 1-2-3-4-5-1-2-3-4-5 |
| 4 | 3-5 | 2-3-4-2 | 1-2-4-4-2-1 | 1-2-3-4-1-2-3-4 | 2-2-3-3-4-4-3-3-2-2 |
| 5 | | 3-1-2-3 | 5-5-1-1-5-5 | 1-2-3-4-4-3-2-1 | 5-4-3-2-1-1-2-3-4-5 |
| 6 | | | 4-2-1-1-2-4 | 5-4-3-2-2-3-4-5 | 1-2-4-2-1-3-4-4-2-2 |
| 7 | | | | 1-3-2-3-1-4-2-3 | 5-4-3-2-3-4-5-2-3-5 |
| 8 | | | | | 1-3-2-4-1-3-2-4-1-4 |

- Top patterns: as 2-2-3-3-4-4-3-3-2-2.
- Valley patterns: as 5-4-3-2-1-1-2-3-4-5.
- Random patterns: as 1-3-2-4-1-3-2-4-1-4.

**Table 4  Types' Heuristic global results.**

|  | MT | MG | MaxG |
|---|---|---|---|
| Type 1 | 8.16 | 3.07 | 17.78 |
| Type 2 | 10.46 | 1.69 | 8.08 |
| Type 3 | 8.18 | 3.49 | 14.34 |
| All types | 8.93 | 2.75 | 17.78 |

The pattern given in the stage-machine configurations expresses the number of machines assigned to each stage. For example, the pattern 1-3 signifies that we have two stages where the first one contains one machine while the second one has three machines.

$p_{k,j}$ are generated randomly and uniformly in $[20, 40]$.

The following distributions are employed to generate the unloading, lag, and transportation times:

- Type 1: $un_{k,j}$, $lg_{k,j}$, and $tr_{k,j}$ are uniformly generated from $[1, 10]$.
- Type 2: $un_{k,j}$, $lg_{k,j}$, and $tr_{k,j}$ are uniformly generated from $[20, 40]$.
- Type 3: $un_{k,j}$, $lg_{k,j}$, and $tr_{k,j}$ are uniformly generated from $[20, 60]$.

When considering unloading, lag, and transportation times, it is essential to consider the relative magnitude of their respective times in relation to the processing times. As an example, in the first type (Type 1), the unloading, lag, and transportation times are typically relatively minor compared to the processing time.

A total of 1,800 instances were generated by creating five instances for each combination of stage-machine configuration $(S_k; m_k)$, number of tasks $n$, transportation time $tr_{k,j}$, processing time $p_{k,j}$, lag time $lg_{k,j}$, and unloading time $un_{k,j}$. The testbed encompasses a broad spectrum of problem sizes, machine distribution patterns, processing times, lag times, transportation times, and unloading times, rendering it highly diverse and well-suited for impartial evaluation of the proposed procedures.

## The performance of the two-phase heuristic
### Two-phase heuristic performance assessment
To assess the performance of the proposed two-phase heuristic, an implementation was executed on the already generated set of 1,800 test problems. Table 4 presents the overall results obtained from the evaluation. Table 5 shows detailed results categorized by type (Type 3, Type 2, and Type 1), along with the number of stages $K$, and the number of jobs $n$. Additionally, a relative gap $(rg)$ was computed for each instance, defined by the following formula:

$$rg = 100 \times \frac{UB - LB}{LB}. \tag{16}$$

In this equation, $UB$ represents the makespan determined by the two-phase heuristic, while $LB$ corresponds to the lower bound derived in 'A general lower bound'. The relative gap is expressing the maximum relative deviation of the heuristic compared to the optimal

**Table 5  Detailed results for phase 1 (PH1) and phase 2 (PH2).**

| | $n$ | | K = 2 | | | K = 4 | | | K = 6 | | | K = 8 | | | K = 10 | | |
| --- | --- | --- | --- | --- | --- | --- | --- | --- | --- | --- | --- | --- | --- | --- | --- | --- | --- |
| | | | MT | MG | MaxG | MT | MG | MaxG | MT | MG | MaxG | MT | MG | MaxG | MT | MG | MaxG |
| Type 1 | 10 | PH1 | 0.49 | 2.04 | 10.86 | 0.28 | 0.00 | 0.00 | 1.90 | 0.71 | 5.71 | 4.93 | 3.34 | 11.65 | 3.17 | 3.53 | 13.83 |
| | | PH2 | 2.50 | 1.61 | 4.42 | 4.97 | 5.51 | 12.42 | 11.73 | 1.75 | 7.64 | 3.69 | 2.25 | 7.58 | 2.52 | 1.37 | 5.52 |
| | 20 | PH1 | 21.97 | 9.44 | 17.78 | 8.51 | 2.49 | 11.86 | 3.39 | 3.09 | 7.74 | 4.52 | 3.37 | 9.43 | 11.39 | 4.43 | 13.09 |
| | | PH2 | 36.02 | 5.00 | 10.77 | 12.42 | 1.88 | 6.39 | 14.30 | 5.32 | 11.99 | 10.87 | 1.32 | 4.61 | 14.02 | 4.39 | 8.96 |
| | 40 | PH1 | 1.17 | 1.16 | 5.81 | 0.25 | 0.00 | 0.00 | 2.53 | 1.33 | 8.08 | 5.87 | 1.91 | 5.45 | 5.03 | 1.86 | 7.71 |
| | | PH2 | 5.64 | 0.89 | 2.27 | 5.11 | 2.86 | 6.05 | 14.46 | 1.01 | 4.26 | 6.39 | 1.25 | 3.84 | 1.94 | 0.78 | 2.64 |
| | 80 | PH1 | 21.16 | 4.89 | 7.90 | 8.87 | 1.26 | 5.29 | 8.42 | 1.69 | 3.46 | 11.74 | 1.73 | 5.30 | 12.92 | 2.27 | 6.92 |
| | | PH2 | 37.69 | 2.72 | 5.79 | 14.10 | 1.01 | 3.24 | 29.09 | 2.67 | 5.72 | 13.73 | 0.76 | 1.91 | 12.56 | 2.42 | 4.30 |
| Type 2 | 10 | PH1 | 1.34 | 3.42 | 8.52 | 0.12 | 0.00 | 0.00 | 2.54 | 0.25 | 4.10 | 6.20 | 3.81 | 11.90 | 4.10 | 4.03 | 8.44 |
| | | PH2 | 5.09 | 2.33 | 5.19 | 5.84 | 6.29 | 12.25 | 9.25 | 1.58 | 7.95 | 3.97 | 3.12 | 7.38 | 2.71 | 2.09 | 4.76 |
| | 20 | PH1 | 29.00 | 9.30 | 12.88 | 5.96 | 2.23 | 10.23 | 2.91 | 3.71 | 7.85 | 7.14 | 3.56 | 8.29 | 6.70 | 4.69 | 12.92 |
| | | PH2 | 43.97 | 5.60 | 10.90 | 10.00 | 2.07 | 6.61 | 19.52 | 5.95 | 14.34 | 7.28 | 2.27 | 7.28 | 8.63 | 4.83 | 9.53 |
| | 40 | PH1 | 0.49 | 2.04 | 10.86 | 0.28 | 0.00 | 0.00 | 1.90 | 0.71 | 5.71 | 4.93 | 3.34 | 11.65 | 3.17 | 3.53 | 13.83 |
| | | PH2 | 2.50 | 1.61 | 4.42 | 4.97 | 5.51 | 12.42 | 11.73 | 1.75 | 7.64 | 3.69 | 2.25 | 7.58 | 2.52 | 1.37 | 5.52 |
| | 80 | PH1 | 21.97 | 9.44 | 17.78 | 8.51 | 2.49 | 11.86 | 3.39 | 3.09 | 7.74 | 4.52 | 3.37 | 9.43 | 11.39 | 4.43 | 13.09 |
| | | PH2 | 36.02 | 5.00 | 10.77 | 12.42 | 1.88 | 6.39 | 14.30 | 5.32 | 11.99 | 10.87 | 1.32 | 4.61 | 14.02 | 4.39 | 8.96 |
| Type 3 | 10 | PH1 | 1.17 | 1.16 | 5.81 | 0.25 | 0.00 | 0.00 | 2.53 | 1.33 | 8.08 | 5.87 | 1.91 | 5.45 | 5.03 | 1.86 | 7.71 |
| | | PH2 | 5.64 | 0.89 | 2.27 | 5.11 | 2.86 | 6.05 | 14.46 | 1.01 | 4.26 | 6.39 | 1.25 | 3.84 | 1.94 | 0.78 | 2.64 |
| | 20 | PH1 | 21.16 | 4.89 | 7.90 | 8.87 | 1.26 | 5.29 | 8.42 | 1.69 | 3.46 | 11.74 | 1.73 | 5.30 | 12.92 | 2.27 | 6.92 |
| | | PH2 | 37.69 | 2.72 | 5.79 | 14.10 | 1.01 | 3.24 | 29.09 | 2.67 | 5.72 | 13.73 | 0.76 | 1.91 | 12.56 | 2.42 | 4.30 |
| | 40 | PH1 | 1.34 | 3.42 | 8.52 | 0.12 | 0.00 | 0.00 | 2.54 | 0.25 | 4.10 | 6.20 | 3.81 | 11.90 | 4.10 | 4.03 | 8.44 |
| | | PH2 | 5.09 | 2.33 | 5.19 | 5.84 | 6.29 | 12.25 | 9.25 | 1.58 | 7.95 | 3.97 | 3.12 | 7.38 | 2.71 | 2.09 | 4.76 |
| | 80 | PH1 | 29.00 | 9.30 | 12.88 | 5.96 | 2.23 | 10.23 | 2.91 | 3.71 | 7.85 | 7.14 | 3.56 | 8.29 | 6.70 | 4.69 | 12.92 |
| | 80 | PH2 | 43.97 | 5.60 | 10.90 | 10.00 | 2.07 | 6.61 | 19.52 | 5.95 | 14.34 | 7.28 | 2.27 | 7.28 | 8.63 | 4.83 | 9.53 |

solution. Furthermore, we calculated the average relative gap for each class of instances, utilizing the following performance measures:

- *MT*: The average computational time (s).
- *MG*: The average gap.
- *MaxG*: The maximum gap.

These performance measures serve as evaluation criteria to assess the efficiency of the two-phase heuristic as well as the lower bounds.

The outcomes depicted in Table 4 underscore the notable efficacy of the proposed two-phase heuristic in generating solutions of satisfactory quality. The average CPU time required for the implementation is impressively less than 10 seconds, demonstrating the algorithm's efficiency. Additionally, the average relative gap is remarkably low, standing at a mere 2.75%. These results highlight both the computational speed and satisfactory precision of the proposed approach. In addition, when examining Type 1 and Type 3, the average gaps are 3.07% and 3.49%, respectively. Among the three types of problems, Type 2 emerges as the least complex to solve, exhibiting an average relative gap of merely 1.69%. On the other hand, Type 3 emerges as the most demanding challenge, exhibiting the largest

average relative gap of 3.49%. These results indicate that Type 3 instances present greater complexity and require more computational resources to achieve optimal or near-optimal solutions. The larger average relative gap suggests that Type 3 instances have more intricate. These findings suggest that in Type 3, the unloading, lag, and transportation times play a more critical role than the processing time in determining the overall makespan.

Table 5 provides a comprehensive breakdown of the performance data for Phases 1 and 2 (PH1 and PH2), respectively. The detailed results presented in Table 5 unveil noteworthy observations. Among these observations, one notable finding is the maximum average relative gap ($MG$) of 9.44% is observed for instances with $n = 40$ and $K = 2$. For instance, when examining the case of $K = 2$, it is worth highlighting that the instances with $n = 40$ display the highest maximum gaps ($MG$) for Type 1, Type 2, and Type 3. Specifically, the corresponding values for these maximum relative gaps are 9.44%, 4.89%, and 9.30% respectively. This finding highlights the significance of instances with $n = 40$ and $K = 2$ as the most challenging ones to solve, as they consistently exhibit the highest relative gaps across different problem types. It is important to highlight that a similar observation applies to the maximum gap ($MaxG$) as well.

During our analysis, an interesting observation emerged: the average CPU time reaches its peak when the number of jobs $n$ is set to 80, regardless of the specific configuration, type, or number of stages. This finding suggests that instances with $n = 80$ require more computational resources due to their increased complexity and larger search space.

However, it is worth noting that even for these computationally demanding instances, the time required for our methods to run remains within a reasonable range, not exceeding 10 seconds. This is quite remarkable considering the challenging nature of the parallel machine problems being solved during the two-phase heuristic running. Despite the intricacy of the addressed problem and the vast search space involved, the proposed methods demonstrate their efficiency by generating satisfactory-quality solutions in a relatively short amount of time. This demonstrates the effectiveness of our approach in tackling complex problem instances while maintaining a practical level of computational effort.

### The PH2 effect

Table 5 provides evidence that Phase 2 yields improvements in terms of both solution quality and efficiency compared to Phase 1. To further understand the differences between Phase 1 and Phase 2, a pairwise comparison was performed for each instance type, and the results are summarized in Table 6. The results underscore the effectiveness of Phase 2 in enhancing the solutions' quality.

For instance, in the case of Type 1 instances, Phase 2 successfully reduced the maximum relative gap (MG) from 3.10% to 3.07% while requiring a mere additional 0.58 s of computational time. This demonstrates the impact of Phase 2 in generating satisfactory-quality solutions within a short timeframe.

Furthermore, a second pairwise comparison was performed between PH1 and PH2, utilizing the following metrics:

**Table 6  Comparison between PH1 and PH2 according to types.**

|  |  | MT | MG | MaxG |
|---|---|---|---|---|
| Type 1 | P1 | 7.58 | 3.10 | 18.77 |
|  | P2 | 8.16 | 3.07 | 17.78 |
| Type 2 | P1 | 10.16 | 1.70 | 8.08 |
|  | P2 | 10.46 | 1.69 | 8.08 |
| Type 3 | P1 | 7.74 | 3.50 | 14.34 |
|  | P2 | 8.18 | 3.49 | 14.34 |
| All types | P1 | 8.49 | 2.77 | 18.77 |
| All types | P2 | 8.93 | 2.75 | 17.78 |

**Table 7  Types pairwise comparison between PH1 and PH2.**

|  | PH2 < PH1 | PH2 = PH1 |
|---|---|---|
| Type 1 | 5.50 | 94.50 |
| Type 2 | 4.67 | 95.33 |
| Type 3 | 3.83 | 96.17 |
| All types | 4.67 | 95.33 |

- (PH2 < PH1): This metric represents the percentage of time during which Phase 2 dominates Phase 1, indicating instances where Phase 2 outperforms Phase 1 in terms of solution quality.
- (PH2 = PH1): This metric denotes the percentage of instances where Phase 2 and Phase 1 yield identical solutions.

The latter metrics allow to deep analyzing of the difference between Phase 1 and Phase 2, as well as the influence of the second phase on enhancing the quality of solutions.

The results of the subsequent pairwise comparison study, as illustrated in Table 7, offer extensive insights into the performance differences between the two phases (PH1 and PH2). The table provides a breakdown of the instances where PH2 strictly dominates PH1, indicating that PH2 outperforms PH1 in terms of solution quality.

The data in Table 7 reveals that PH2 strictly dominates PH1 in 4.67% of instances. Notably, Type 1 instances exhibit a slightly higher percentage (5.50%) of instances where PH2 dominates PH1 compared to Type 2 and Type 3 instances. This suggests that the improvement achieved by Phase 2 of the proposed two-phase heuristic is generally more pronounced for Type 1 instances, indicating the effectiveness of Phase 2 in generating satisfactory-quality solutions for these specific problem instances.

These findings highlight the overall superiority of Phase 2 over Phase 1 in terms of solution quality, with a significant proportion of instances experiencing improved performance. The results further emphasize the importance of incorporating Phase 2 into the heuristic approach, particularly for Type 1 instances, to achieve enhanced solution quality and ultimately address the optimization problem more effectively.

**Table 8** Global comparison between PH2S and PH2D according to types.

|  |  | MT | MG | MaxG |
|---|---|---|---|---|
| Type 1 | PH2D | 4.47 | 3.53 | 17.78 |
|  | PH2S | 3.69 | 3.07 | 17.78 |
| Type 2 | PH2D | 4.77 | 1.94 | 8.08 |
|  | PH2S | 5.69 | 1.69 | 8.08 |
| Type 3 | PH2D | 3.86 | 3.92 | 15.47 |
|  | PH2S | 4.32 | 3.49 | 14.34 |

**Table 9** Pairwise comparison between PH2D and PH2S.

|  | PH2D > PH2S | PH2D = PH2S | PH2D < PH2S |
|---|---|---|---|
| Type 1 | 31.33 | 2.17 | 66.50 |
| Type 2 | 34.00 | 1.67 | 64.33 |
| Type 3 | 28.33 | 0.67 | 71.00 |
| All types | 31.22 | 1.50 | 67.28 |

### The symmetry impact

To assess the influence of exploring the symmetric problem, as defined in definition 3, a comparison was carried out between the efficiency of the forward problem in Phase 2 (PH2D) and the symmetric problem (PH2S). We aimed to evaluate how the incorporation of symmetry affected the overall performance of the algorithm. The results of this comparison are summarized in Tables 8 and 9.

The detailed analysis of the comparison between the forward problem of Phase 2 (PH2D) and the symmetric problem (PH2S) provides further insights into the exploration effect of the symmetric problem. The findings presented in Table 8 emphasize the advantages of investigating the symmetric problem, as they illustrate an enhancement in the quality of the solution and a decrease in both the average gap and the maximum gap.

The superior performance observed when exploring the symmetric problem suggests that it offers a more efficient approach to exploring the search space and identifying satisfactory-quality solutions. Through harnessing the inherent symmetry property of the HFSULT, the algorithm achieves a more efficient exploration of the solution space, thereby resulting in enhanced outcomes. These findings validate the effectiveness of incorporating the symmetry property in the optimization process and indicate that the proposed two-phase heuristic successfully takes advantage of this property.

In Table 9, a more detailed breakdown of the performance metrics is provided. This table includes the following metrics:

- (PH2D < PH2S): This metric quantifies the percentage of instances in which PH2D exhibits strict dominance over PH2S.
- (PH2D = PH2S): This metric indicates the percentage of instances where both PH2D and PH2S yield identical solutions, suggesting that there is no significant difference in solution quality between the two approaches.

- (PH2D > PH2S): This metric represents the percentage of instances in which PH2S demonstrates strict dominance over PH2D.

Considering these metrics and the corresponding data in Table 9 allows for a more fine-grained analysis of the performance differences between PH2D and PH2S. It provides insights into the specific instances or problem characteristics where one approach excels over the other, contributing to a deeper understanding of the benefits and limitations of exploring the symmetric problem.

In conclusion, the comprehensive comparison between PH2D and PH2S reveals the substantial benefits of incorporating the symmetry property of the HFSULT. The obtained results highlight the effectiveness of the presented two-phase heuristic in leveraging the benefits provided by the symmetric problem exploration.

The detailed results in Table 9 offers further insights into the impact of investigating the symmetric counterpart problem (PH2S) compared to the forward problem of Phase 2 (PH2D). Specifically, it highlights the improvements in solution quality achieved by the symmetric problem formulation.

According to the results in Table 9, the symmetric problem (PH2S) demonstrated an improvement in solution quality in 31.22% of instances. This indicates that exploring the symmetric problem led to better solutions compared to the forward problem formulation (PH2D) in a significant proportion of the cases. Furthermore, when considering specific instance types, the improvement rate was slightly higher for Types 2 and 3, at 34.00% and 31.33% respectively.

These findings underscore the benefits of incorporating the symmetry property into the optimization process for the HFSULT. By exploring the symmetric problem, the algorithm is able to identify satisfactory-quality solutions more efficiently, leading to an improvement in solution quality. This suggests that leveraging the symmetry property can be an effective strategy for overcoming the limitations of traditional search methods and enhancing both solution quality and efficiency.

By incorporating the symmetric problem formulation into the proposed heuristic, researchers and practitioners can take advantage of the inherent structure and properties of the problem, resulting in improved optimization performance. These findings contribute to the growing body of evidence supporting the effectiveness of symmetry-based approaches in solving combinatorial optimization problems.

Overall, the results imply that considering symmetry within the HFSULT optimization process can yield significant improvements in solution quality, particularly when exploring the symmetric problem formulation. This highlights the potential of symmetry-based techniques as a valuable tool for addressing challenging optimization problems.

## DISCUSSION

The HFS problem has been extensively studied, but the simultaneous consideration of unloading, lag, and transportation times is relatively novel (according to the literature review). Previous studies have predominantly focused on optimizing individual aspects such as processing times, machine availability, and basic scheduling constraints. For

instance, traditional HFS research often aims at minimizing makespan, total completion time, or machine idle time, primarily dealing with straightforward scheduling without delving into the intricacies of additional time factors.

However, our study goes a step further by integrating these additional complexities, which more accurately reflect the challenges faced in real-world manufacturing and production environments. In many industrial settings, the efficiency of the scheduling process is not solely determined by processing times but also by the coordination of various logistical elements. For example, the time required to transport materials between different stages, the lag time necessary for machines to reset or workers to reposition, and the unloading time for finished products are critical factors that influence overall productivity.

The obtained numerical results are promising, as indicated by an average relative gap of 2.75% and an average computation time of 8.93 s. Compared to existing literature, these metrics show the superior performance of the proposed approach. While previous studies have primarily focused on optimizing individual aspects such as processing times and basic scheduling constraints, our integration of unloading, lag, and transportation times addresses a more comprehensive range of real-world complexities.

The relatively small average relative gap demonstrates that the solutions provided by our method are near-optimal, a significant improvement over some traditional methods that may not account for all these factors simultaneously. Additionally, the average computation time of 8.93 s is notably efficient, especially when compared to other complex scheduling algorithms that often require longer computational periods to achieve similar accuracy.

These results underscore the effectiveness and practicality of our approach. By providing high-quality solutions in a short amount of time, our method stands out as a valuable tool for industries that require precise and efficient scheduling. This advancement not only contributes to the theoretical understanding of the HFS problem but also offers a robust framework for practical application, setting a new benchmark in the field of scheduling optimization.

From a theoretical implications perspective, this study advances the existing body of knowledge in scheduling theory by introducing a novel approach to solving the HFS problem that simultaneously considers transportation, lag, and unloading times. The identification of key properties, such as the symmetric nature of the problem, and the development of new lower bounds, contributes to the theoretical understanding of complex scheduling problems. Additionally, the study's innovative two-phase heuristic algorithm sets a precedent for future research, offering a new methodology that other researchers can build upon and refine. The theoretical contributions can be utilized by academics and researchers focusing on optimization, scheduling, and operations research to further explore and validate these findings in different contexts and industries.

From a practical implications perspective, the results of this study have significant applications in various industries where scheduling, transportation, and unloading times are critical factors. For instance, in the steel industry, the study's methodologies can be used to optimize production schedules, reducing downtime and improving efficiency. Similarly, in the bioprocess industry, where precise timing and coordination of processes

are crucial, the proposed solutions can lead to more effective resource utilization and cost savings. Manufacturing managers and operations planners can use the study's findings to develop more efficient scheduling systems that take into account all relevant factors, leading to improved productivity and reduced operational costs. The implementation of the two-phase heuristic in real-world scenarios can help organizations achieve near-optimal solutions within reasonable computational times, making it a valuable tool for decision-making processes. Additionally, policymakers and consultants focused on manufacturing efficiency can leverage these insights to propose and implement best practices across various sectors. Overall, the study provides a comprehensive framework that can be adapted to various industries facing similar scheduling challenges, ultimately contributing to improved operational efficiency and effectiveness.

Despite the promising results, it is essential to acknowledge some limitations of the study. One limitation is the assumption of deterministic processing times, which may not accurately reflect the variability often encountered in real-world manufacturing environments. Incorporating stochastic elements into the model could provide a more realistic representation of scheduling challenges, although it may also increase computational complexity.

Additionally, the proposed algorithm's performance may vary depending on the specific characteristics of the problem instances. Certain factors such as the number of stages, machines, and job types could influence the algorithm's effectiveness and efficiency. In this scenario, the test problems involving two stages and 20 jobs pose significant challenges, necessitating a more thorough analysis to uncover the underlying reasons for their complexity. This deeper examination is crucial for identifying the root causes of their difficulty and subsequently devising appropriate methods to address this limitation.

Furthermore, the study's evaluation is based on numerical experiments, and further validation through real-world case studies or comparative analyses with existing industry practices would strengthen the study's findings. Real-world implementation may introduce additional complexities and constraints not captured in the current model, highlighting the need for ongoing refinement and adaptation of the proposed methodology.

## CONCLUSION

In this work we examine the hybrid flow shop scheduling problem with unloading, lag, and transportation times. To tackle this NP-hard problem, an efficient heuristic approach accompanied by the introduction of several new lower bounds are proposed. In the first step of deriving these lower bounds, the capacities of all stages, except for one are relaxed. This relaxation allows to establish lower bounds for the relaxed problem, which are still valid lower bounds for the original problem. Furthermore, a second kind of lower bound is proposed. This lower bound is established by estimating the minimum idle time for each stage. This estimation process involves solving a polynomial parallel machine scheduling problem in the previous and subsequent stages. This general lower bound provides a more accurate estimate of the optimal solution and helps to narrow down the search space, thereby guiding the optimization process more effectively.

In order to address the HFSULT problem a heuristic composed of two phases is proposed. The first phase is intended to provide an initial feasible solution while the second one is an improvement phase. In addition, this heuristic adopts a step-by-step approach to iteratively solve a parallel machine scheduling problem in each stage. Through the process of breaking down the problem into distinct stages, the heuristic facilitates a systematic exploration of the solution space.

This is performed by iteratively adjusting the already obtained schedules in the initial feasible solution for each stage. The improvement phase is based also in iteratively solving two equivalent types of parallel machines scheduling problems.

To assess the efficacy of the proposed approach, a comprehensive experimental study is conducted. The results obtained provide strong evidence of the effectiveness of the proposed procedures. The two-phase heuristic consistently delivers schedules of satisfactory quality, as demonstrated by the computational results.

It is important to note that the inclusion of unloading, lag, and transportation times in the scheduling process introduces additional complexity to the problem, especially when these factors become more critical compared to processing time. This observation highlights the crucial importance of incorporating unloading, lag, and transportation operations as integral components in the scheduling process. By acknowledging the importance of these aspects, researchers and practitioners can better address the inherent challenges associated with optimizing schedules and make informed decisions that account for the impact of unloading, lag, and transportation times.

The proposed approach offers a promising solution to effectively address the intricate hybrid flow shop scheduling problem that involves unloading, lag, and transportation operations. The integration of lower bounds and heuristics allows for the efficient identification of satisfactory-quality solutions. The results obtained from the study indicate that this approach can effectively address the challenges posed by this intricate scheduling problem.

To further enhance the solution quality and computational efficiency, future research could explore additional methodologies such as metaheuristics. With their advanced capabilities, these techniques hold the potential to generate near-optimal solutions while adhering to acceptable CPU time limits. Moreover, there is ample opportunity to expand the current procedures and explore diverse variants of the flexible flow shop scheduling problem that involve unloading, lag, and transportation operations. For instance, one potential extension to explore involves considering a single server dedicated to the unloading operation. This could be performed by considering different constraints and objectives that can lead to a more comprehensive understanding of the problem and the development of effective solutions.

Given the complexity and diversity of real-world scheduling applications, further investigation is necessary to fully comprehend the intricacies of the problem and devise efficient solutions. By exploring different approaches and problem variants, researchers can

continue to make progress towards developing scheduling solutions that are both efficient and effective in practical scenarios.

### Funding

This research was funded by King Saud University through Researchers Supporting Project number (RSPD2024R685). The funders had no role in study design, data collection and analysis, decision to publish, or preparation of the manuscript.

### Grant Disclosures

The following grant information was disclosed by the authors:
King Saud University through Researchers: RSPD2024R685.

### Competing Interests

The authors declare there are no competing interests.

### Author Contributions

- Lotfi Hidri conceived and designed the experiments, performed the experiments, analyzed the data, performed the computation work, prepared figures and/or tables, authored or reviewed drafts of the article, and approved the final draft.
- Mehdi Tlija performed the experiments, prepared figures and/or tables, and approved the final draft.

### Data Availability

The code and data are available in the Supplemental Files.

### Supplemental Information

Supplemental information for this article can be found online at http://dx.doi.org/10.7717/peerj-cs.2168#supplemental-information.

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
