# Peer review of "Multi-stage hybrid flow shop scheduling problem with lag, unloading, and transportation times"

_PeerJ Computer Science, doi:10.7717/peerj-cs.2168_

## Round 0.1 · original submission · Major Revisions

Dear author,

Your paper has been reviewed by three reviewers and we are happy to allow you to revise the paper. Please mark all changes in the paper and provide cover letter with replies point to point.

**Language Note:** PeerJ staff have identified that the English language needs to be improved. When you prepare your next revision, please either (i) have a colleague who is proficient in English and familiar with the subject matter review your manuscript, or (ii) contact a professional editing service to review your manuscript. PeerJ can provide language editing services - you can contact us at [email protected] for pricing (be sure to provide your manuscript number and title). – PeerJ Staff

Reviewer 1 ·

Basic reporting

- English writing is acceptable. None serious issues were detected.
- All references cited in the paper must be in the reference list, and vice versa. For example, the reference „(Vandevelde et al., 2005)“ is not listed in the reference list, and the reference „Botta-Genoulaz V. 2000a. Hybrid flow shop scheduling with precedence constraints and time lags to minimize maximum lateness. International Journal of Production Economics 64:101-895.“ is not cited anywhere in the paper. Check all references.
- Some references from the reference list are not complete. They are missing information such as volume numbers, DOI numbers, etc. Complete all references.
- Some sections of the paper are too partitioned. It seems like there is no need to introduce some of the sub-headings. Some sub-sections contain only a few sentences.

Experimental design

- The subject of the paper is interesting and in line with the aims and scope of the Journal.
- Research questions are well defined and paper is well-written.
- The abstract should be improved. It seems repetitive. Try to better articulate the background, aim of the study, methodology with contributions, main results, and conclusions.
- The Introduction is too extensive and includes a literature review. I think this section should be separated into two separate sections, Introduction and Literature review.
- It is unclear whether the proposed heuristics are developed by the authors or based on some previous studies. If the second is the case, the authors should include these previous studies in the literature review.
- The notation is very extensive. The authors should check once again if everything is in order, i.e. to make sure they are not using the same notation for different variables or values.

Validity of the findings

- The paper does not have a proper discussion. The authors did not discuss how the results can be interpreted from the perspective of previous studies. Discussion should clearly and concisely explain the significance of the obtained results to demonstrate the actual contribution of the article to this field of research when compared with the existing and studied literature.
- The discussion should point out the theoretical and practical implications of the study. Who and for what can use the results of this study?
- The authors should elaborate on the potential limitations of the study.

Additional comments

- There are some typos (e.g. in line 302). Check and correct similar errors in the entire paper.ž
- Tables are not uniformly formatted.
- Some equations are not numbered.

Reviewer 2 ·

Basic reporting

• The spelling of this reference "(NARASTMHAN & PANWALKAR 1984)" on line 55 is not appropriate. It should be corrected.

• Authors should also review the following articles on hybrid flow type scheduling problems with multi-processor tasks.

- “A new memetic global and local search algorithm for solving hybrid flow shop with multiprocessor task scheduling problem”
- “Hybrid flow shop with multiprocessor task scheduling based on earliness and tardiness penalties”
- “Multiprocessor task scheduling in multistage hybrid flow-shops: A parallel greedy algorithm approach”

• The sentence on line 112 “In some scheduling problems…” is rewrite.

• The manuscript is not clear. It should be redesigned. The hybrid flow shop with multiprocessor literature should be reviewed.

Experimental design

The research is convenient the scope of the journal. But,

• The main contribution of the article is not understood, it is confused.
• The introduction section is confused. The literature review is confused. It should be clear.
• The introduction section should be redesigned.

Validity of the findings

• The example 1 on line 302 is not clear. How the feasible solution obtained should be explained.

• The formulation on line 418 is not understandable. It should be explained by their notations.

• The formulation on line 457 is not understandable. It should be explained by their notations.

• The proposed hybrid solution method is not clear. The proposed methods should be explained in detailed.

Reviewer 3 ·

Basic reporting

no comment

Experimental design

no comment

Validity of the findings

no comment

Additional comments

The topic is very popular. Many problems in everyday life are optimization problems. The
scheduling problem appears very often in practice in many different ways. When we talk
about it, then, in general sense, we talk about the allocation of resources in such a way as to
optimize one or more objective functions.
Today, it is unthinkable to solve complex scheduling problems without applying exact or
nearly exact optimization methods. Research results indicate that this approach can solve
these problems.
Experimental results provide strong evidence of the effectiveness of the proposed procedure.
The two-phase heuristics consistently delivers schedules of satisfactory quality. It is very
significant that the authors include unloading, delay and transportation times in the planning
process which adds additional complexity to the problem, especially when these factors
collide with processing time.

---

## Round 0.2 · accepted · Accept

Dear authors,

Your paper has been reviewed again and the reviewer proposed acceptance of your paper. Congratulations.

Reviewer 1 ·

Basic reporting

The authors have successfully addressed all issues from the previous review round.

Experimental design

The authors have successfully addressed all issues from the previous review round.

Validity of the findings

The authors have successfully addressed all issues from the previous review round.

Additional comments

The authors have successfully addressed all issues from the previous review round.